

# The method ADAMONT v1.0 for statistical adjustment of climate projections applicable to energy balance land surface models

Deborah Verfaillie[1], Michel Déqué[1], Samuel Morin[1], and Matthieu Lafaysse[1]

[1]CNRM UMR 3589, Météo-France/CNRS, Toulouse, France

*Correspondence to:* Deborah Verfaillie (deborah.verfaillie@meteo.fr)

**Abstract.**

We introduce the method ADAMONT v1.0 to adjust and disaggregate daily climate projections from a regional climate model against an observational dataset at hourly time resolution. The method uses a refined quantile mapping approach for statistical adjustment and an analogous method for sub-

daily disaggregation. The method produces ultimately adjusted hourly time series of temperature, precipitation, wind speed, humidity, and short- and longwave radiation, which can in turn be used to force any energy balance land surface model. While the method is generic and can be employed on any appropriate observation time series, here we focus on the description and evaluation of the method in the French mountainous regions. The observational dataset used here is the SAFRAN me-

teorological reanalysis, which covers the entire French Alps split into 23 massifs, within which meteorological conditions are provided for several 300 m elevation bands. In order to evaluate the skills of the method itself, it is applied to the ALADIN-Climate v5 RCM using the ERA-Interim reanalysis as boundary conditions, for the time period from 1980 to 2010. Results of the ADAMONT method are compared to the SAFRAN reanalysis itself. Various evaluation criteria are used for tempera-

ture, precipitation, but also snow depth, which is computed by the SURFEX/ISBA-Crocus model using the meteorological driving data from either the adjusted RCM data, or the SAFRAN reanaly-sis itself. The evaluation addresses in particular the time transferability of the method (using various learning/application time periods), the impact of the RCM grid point selection procedure for each massif/altitude band configuration, and the inter-variable consistency of the adjusted meteorological

data generated by the method. Results show that the performance of the method is satisfactory, with similar or even better evaluation metrics than alternative methods. However, results for air tempera-ture are generally better than for precipitation. Results in terms of snow depth are satisfactory, which can be viewed as indicating a reasonably good inter-variable consistency of the meteorological data produced by the method. In terms of temporal transferability (evaluated over time periods of 15 years

only), results depend on the learning period. In terms of RCM grid point selection technique, the use of a complex RCM grid points selection technique, taking into account horizontal but also altitudinal





proximity to SAFRAN massif centre points/altitude couples, generally degrades evaluation metrics for high altitudes, compared to a simpler grid point selection method based on horizontal distance.

## 1 Introduction

Projections of future climate change in terms of meteorological conditions and their impacts are requested for many scientific and societal applications (IPCC, 2013, 2014a, b, c). For a given socio-economic or greenhouse-gas concentration scenario, these projections generally concern future temperature and precipitation, and associated extreme events, and are usually generated using the outputs of global climate models (GCMs) and regional climate models (RCMs). However, GCMs and RCMs

suffer from biases compared to local observations (Christensen et al., 2008; Rauscher et al., 2010; Kotlarski et al., 2014). Raw climate projections must therefore be adjusted (Déqué, 2007; Themeßl et al., 2011; Gobiet et al., 2015), before they can be used as such (meteorological conditions), or in order to drive specific impact models. Various downscaling and adjustment methods have been developed (Maraun et al., 2010; Teutschbein and Seibert, 2012, 2013). They all require an obser-

vation dataset which (i) meets the data requirements of the application and (ii) is sufficiently long and reliable to be used to infer the relationships between the observations and the raw climate projections during the observation time period. Several approaches, such as the analog method, search for relationships between observed large-scale predictors (generally from reanalyses) and observed local-scale predictands (Vrac et al., 2007a; Dayon et al., 2015). In contrast, model output statistics

approaches calibrate model outputs against observations, with various levels of complexity, such as scaling methods (linear scaling, local intensity scaling, variance scaling, ...), delta-change methods (e.g., Abegg et al., 2007; Hantel and Hirtl-Wielke, 2007; Schmucki et al., 2014) and distribution mapping methods (e.g., Boe et al., 2007; Déqué, 2007; Gobiet et al., 2015; Olsson et al., 2015). The latter include quantile mapping, which is considered as an efficient and easy to implement ad-

justment method (Themeßl et al., 2011; Teutschbein and Seibert, 2012; Maurer and Pierce, 2014; Gobiet et al., 2015). The main advantage of this method is that it adjusts deviations in the shape of the distribution, and is thus able to adjust deviations not only for the mean but the entire probability distribution function (Themeßl et al., 2011). Moreover, the adjustment is not strictly restricted to the range of observed values in the reference period, which is the case for example for methods based

on analog weather patterns (e.g., Déqué, 2007; Themeßl et al., 2011; Rousselot et al., 2012; Dayon et al., 2015), provided that values based on the lowermost and uppermost quantiles are handled appropriately (Gobiet et al., 2015). It can thus be used for evaluation of climate extremes or projections at the end of the 21st century, as long as the probability associated with these events is robustly estimated from a long enough sample. The main limits of quantile mapping are the assumption of

time-invariant model deviation to observations on which it is based, and the fact that the temporal properties of the model are not adjusted. If the model has a chronological behaviour which differs




from the observations (too chaotic or too persistent), this will not be adjusted (Déqué, 2007). More-
over, quantile mapping does not guarantee the spatial and inter-variable consistency, in contrast to
e.g. the analog method.

Climate projections in mountainous regions, which are motivated by a broad range of geophysi-
cal, environmental and societally relevant scientific challenges (Martin et al., 1994; Beniston, 1997;
Jomelli et al., 2009; Castebrunet et al., 2014; Piazza et al., 2014; Schmucki et al., 2014; Lafaysse
et al., 2014; Boulangeat et al., 2014; Thuiller et al., 2014; Castebrunet et al., 2014; Francois et al.,
2015; Spandre et al., 2016), are particularly sensitive to the quality of the adjustment method. In-
deed, regional climate model resolutions typically between 10 and 50 km are not sufficient to capture
the fine-scale processes and thresholds at play. Resolving altitude dependencies is critical, espe-
cially for snow-related issues (because of the temperature dependency of the snow/rain transition).
Furthermore, not only temperature and precipitation act on the snowpack, but a broader range of
meteorological conditions and their diurnal variations. As a consequence, considering only adjusted
daily temperature and precipitation would miss some of the non-linear response of the snowpack.
Such phenomena cannot be addressed using delta-change methods, which by definition apply fixed
changes to an observed time series, conserving its statistical persistence properties and seasonality
(e.g., Abegg et al., 2007; Hantel and Hirtl-Wielke, 2007; Schmucki et al., 2014; Marty et al., 2017)
although those could evolve significantly under changed climate conditions.

Here we introduce the ADAMONT v1.0 method, to adjust climate model projections in order
to provide hourly adjusted meteorological conditions for past and future conditions based on cli-
mate model output and observational datasets. Although it could be applied for GCM output, it
was primarily designed to process RCM output. Indeed, raw regional climate projection data are
increasingly made available, e.g. the World Climate Research Program (WCRP) Coordinated Re-
gional Downscaling Experiment (CORDEX, Giorgi et al. (2009)), whose aim is to improve and dis-
tribute regional climate modelling worldwide. Its European branch, EURO-CORDEX (Jacob et al.,
2014), gathers regional climate simulations over Europe from 30 different modelling groups at 50
km (EUR-44) and 12.5 km (EUR-11) resolution. On the observation side, the use of surface mete-
orological reanalysis is a powerful alternative to station observation data to provide the necessary
observational dataset (Berg et al., 2015). Indeed, the process by which such reanalyses are generated
addresses the time and space variations of the meteorological conditions, and by design they consist
of gap-free and complete time series. Here we describe the use of the ADAMONT method based on
RCM model output comparable to EURO-CORDEX and on the mountain meteorological reanalysis
SAFRAN. SAFRAN was developed specifically to address the needs of snowpack numerical simu-
lations in mountainous regions, and contains hourly time series of temperature, precipitation, wind
speed, humidity, and short- and longwave radiation for so-called massifs (ranging between 500 and
$2,000\,km^2$ in the French Alps) by elevation steps of $300\,m$ (Durand et al., 2009a, b). Here, quan-
tile mapping is applied using daily outputs from a given RCM for all the variables provided in the



SAFRAN reanalysis. Following a subdaily disaggregation step based on analog days selection from
the reanalysis itself, these hourly adjusted fields are then used to force the SURFEX/ISBA-Crocus
(Vionnet et al., 2012) model over the French Alps. We evaluate the performance of the ADAMONT
method, by applying it to the ALADIN-Climate v5 RCM (Colin et al., 2010) forced by the ERA-
Interim reanalysis (Dee et al., 2011) over the period 1980-2010. Sect. 2 describes the models used
and the evaluation approach. Sects. 3 and 4 contain the results and their discussions, respectively,
and general conclusions are drawn in Sect. 5.

## 2 Models and methods

### 2.1 Description of the ADAMONT method

ADAMONT is primarily a quantile mapping adjustment method (Déqué, 2007; Gobiet et al., 2015).
In general, quantile mapping is considered one of the most efficient bias adjustment methods avail-
able (Themeßl et al., 2011; Maurer and Pierce, 2014; Gobiet et al., 2015). It consists in adjusting
the quantiles of the simulated historical distribution based on the quantiles of the observed distribu-
tion. The main issues with quantile mapping relate to the assumption of time-invariant model biases,
the fact that temporal properties of the RCM are untouched by the adjustment method and that the
spatial and inter-variable consistency is not guaranteed. Moreover, Driouech et al. (2009) showed
that for mid-latitude climates, such as that in Morocco, quantile mapping adjustment can vary for
different weather regimes, because model biases vary in different regimes. Similarly, Addor et al.
(2016) demonstrated the sensitivity of quantile mapping adjustment to circulation biases over the
Alpine domain. Additionally, the frequency of weather regimes may change in a changing climate
(Boe et al., 2006; Cattiaux et al., 2013). To improve the stationarity of our method in a changing cli-
mate, weather regimes are thus taken into account in our method, i.e. quantile adjustment functions
are computed and applied depending on the weather regime.

Assuming the availability of a gap-free meteorological observational dataset at hourly time resolu-
tion, and RCM model outputs covering the geographical domain of interest, the statistical adjustment
method ADAMONT consists in the following steps:

1. RCM grid point selection: For each observation point, a RCM grid point is selected, by mini-
   mizing the following distance:

$$\sqrt{(\Delta x)^2 + (\Delta y)^2 + (N \times \Delta z)^2}, \tag{1}$$

   where $\Delta x$, $\Delta y$ and $\Delta z$ represent the longitudinal, latitudinal and vertical distances (in km)
   between the observation point and the RCM grid points, and $N$ is referred to as the elevation
   factor. Values of 0, 50 and 100 were tested, but only results using a value of 0 and 50 (N50)
   are reported in this study. The factor $N$ is a scaling factor between horizontal and vertical
   distances, allowing to take into account the strong dependence of meteorological variables



(mainly precipitation and temperature) on altitude (e.g., Gottardi et al., 2012; Kotlarski et al., 2012).

2. Weather regime computation: Each day of the RCM and observational records are clustered into different daily weather regimes based on the geopotential height at 500 hPa, following Michelangeli et al. (1995), similar to the method described in Driouech et al. (2010). Weather regimes clusters are computed on the basis of a large scale meteorological reanalysis consistent with the observational dataset (in our case, ERA-Interim reanalysis, Dee and Uppala, 2009), and used to infer the weather regime for each date of the RCM dataset based on the synoptic fields of the GCM model used as boundary condition for the RCM. A classifiability and reproducibility analysis performed by Michelangeli et al. (1995) showed that 4 weather regimes can reasonably be chosen for Europe. This number also ensures a sufficiently large size of the datasets for quantile mapping (which are, as described below, further partitioned into 4 seasons DJF, MAM, JJA, SON).

3. Integration from hourly to daily observations: The observational data are integrated from hourly to daily time resolution, depending on the variable considered (see Table 1) : for temperature, the daily (6 am to 6 am the next day) minimum and maximum values are selected, for wind speed and humidity, the last value of each day (at 6 am) is selected (in order to be comparable to an instantaneous value), and for precipitation and radiation, the daily mean is used.

4. Computation of quantile distributions: The quantiles (99 percentiles + 0.5 % and 99.5 % quantiles) of the observational dataset and corresponding RCM grid point distributions are calculated for each variable, each season (DJF, MAM, JJA, SON) and each of the four weather regimes, for a reference (also referred to as learning) time period when both datasets are available.

5. Quantile mapping: Quantile mapping is then applied to the entire RCM dataset for the application time period, taking into account the season and the weather regime. A linear interpolation is used for quantile values between the quantiles values specifically computed (99 percentiles + 0.5 % and 99.5 % quantiles). For RCM values greater than the 99.5 % quantile, a constant adjustment based on the value of this last quantile is applied. For precipitation, it can happen that for low quantiles, the probability of precipitation is lower in the RCM than in the observation dataset (i.e. several null values in the RCM, which can correspond to different positive values in the observational data). In this case, a random draw is performed amongst the observation values within the same quantile.

6. Selection of analogue date for sub-daily disaggregation: For each day in the RCM dataset, an analogous date is chosen in the observational dataset, matching the following criteria: the



month and the weather regime must be the same as in the RCM dataset, and whenever possible, consecutive time slices are chosen in the observational dataset in order to avoid artificial jumps

in the final data linked to the choice of analogues. A further criterion can be applied to ensure that the weather situations are even more comparable between the RCM date and the analogous date from the observational record, based on precipitation consistency (wet vs. dry conditions). A threshold of $1 \, \text{kg m}^{-2} \, \text{day}^{-1}$ on total precipitation is applied to partition dates between dry and wet conditions. For each RCM date, a random draw amongst all available observational

dates is performed, then the dates are browsed through until one meets all the requirements outlined above. This analogous day is then used in the following step for all variables.

7. Sub-daily disaggregation: The adjusted RCM dataset is disaggregated from a daily integration period into an hourly time step by using the hourly observational data from each analogous date chosen in the previous step to reconstruct the daily cycle of the data:

$$X_{RCM}^h(i) = a \times X_{OBS}^h + b, \tag{2}$$

where $X_{RCM}^h(i)$ is the hourly adjusted RCM value of the variable X and $X_{OBS}^h$ is the hourly observational value of the same variable from the chosen analogous date (step 6). Different criteria are chosen to calculate a and b, depending on the variable considered (Table 1). For the disaggregation of RCM adjusted temperature from daily to hourly (Table 1), a compromise

must be made between obtaining minimum and maximum daily values as close as possible to RCM adjusted daily minimum and maximum and minimizing the possible jump in adjusted values between consecutive days. This is achieved by minimising the function:

$$Q(\alpha) = [T_{RCM}^h(1h, i) - T_{RCM}^h(24h, i-1)]^2 + \alpha[Tmin_{RCM}^h(i) - Tmin_{RCM}^{d,adj}(i)]^2$$
$$+ \alpha[Tmax_{RCM}^h(i) - Tmax_{RCM}^{d,adj}(i)]^2, \tag{3}$$

where $T_{RCM}^h(1h, i)$ and $T_{RCM}^h(24h, i-1)$ are the hourly adjusted RCM temperature val-

ues at the first time step of day $i$ and at the last time step of day $i-1$, $Tmin_{RCM}^h(i)$ and $Tmax_{RCM}^h(i)$ are the hourly minimum and maximum adjusted RCM temperature values respectively, and $Tmin_{RCM}^{d,adj}(i)$ and $Tmax_{RCM}^{d,adj}(i)$ are the daily minimum and maximum adjusted RCM temperature values respectively (Fig. 2). $\alpha$ is a parameter which can be tuned to balance the importance of the minimisation of differences between daily and hourly RCM

minima and maxima and the minimisation of the jump between two consecutive days. For a value of $\alpha$ of zero, there would be no jump in values between consecutive days, but the values of $Tmin_{RCM}^h(i)$ and $Tmax_{RCM}^h(i)$ could be far from the values of $Tmin_{RCM}^{d,adj}(i)$ and $Tmax_{RCM}^{d,adj}(i)$. For an infinitely large value of $\alpha$, the minimum and maximum hourly and daily values would match, but the jump between consecutive days could be significant.





Sensitivity tests yielded an optimal value of 2 for $\alpha$. Following eq. 2, eq. 3 transforms into:

$$Q(\alpha, a, b) = [a \times T^h_{OBS}(1h) + b - T^h_{RCM}(24h, i-1)]^2$$
$$+ \alpha[a \times Tmin^h_{OBS} + b - Tmin^{d,adj}_{RCM}(i)]^2$$
$$+ \alpha[a \times Tmax^h_{OBS} + b - Tmax^{d,adj}_{RCM}(i)]^2. \tag{4}$$

By searching for the local minima $\delta Q/\delta a = 0$ and $\delta Q/\delta b = 0$, $a$ and $b$ can be determined, and the hourly adjusted RCM temperature can be obtained following eq. 2. This procedure is only applied for temperature, because the use of the maximum and minimum criterion can

lead to important jumps between consecutive days, which is not the case for other variables (Table 1). For humidity, eq. 2 is solved using $b = 0$ and $a = X^{d,adj}_{RCM}(i)/X^h_{SAF}(24h, i)$, so that the hourly adjusted RCM value and the hourly observational value at the last time step of day i ($X^h_{SAF}(24h, i)$) are equal. For wind speed, the same calculation as for humidity is applied, except if $a > 1$ (i.e., $X^{d,adj}_{RCM}(i) > X^h_{OBS}(24h, i)$). If so, $b = X^{d,adj}_{RCM}(i) - X^h_{SAF}(24h, i)$ is cal-

culated. For humidity and wind speed, if $X^h_{OBS}(24h, i) \leq 10^{-10}$, $a = 0$. For precipitation and radiation, $b = 0$ and $a = X^{d,adj}_{RCM}(i)/X^h_{OBS}(mean, i)$, so that the mean hourly adjusted RCM value and the mean hourly SAFRAN value of day i are equal. For solar radiation, if $X^h_{OBS}(mean, i) \leq 10^{-10}$, $a = 0$. For precipitation, if this is the case, $a = 1$.

8. Snow/rain partitioning: Total precipitation is separated into rainfall and snowfall based on

hourly adjusted temperature (a threshold of 1 °C is used for the transition from snow to rain). As mentioned above, inter-variable consistency is not guaranteed by quantile mapping. Given the importance of the consistency between temperature and precipitation in many applications and in particular in mountainous areas, given that precipitation and temperature are corrected independently from each other (step 5), and because the adjustment can differ for the differ-

ent precipitation phases, the relationship between temperature and precipitation phase may be modified by quantile mapping, so that the adjusted rain and snow distributions may lose consistency. To avoid this, Olsson et al. (2015) separated temperature data into wet and dry days before adjustment. In our case an additional quantile mapping against the observational dataset is applied for daily cumulated adjusted RCM rainfall and snowfall separately. Hourly

adjusted RCM rainfall and snowfall ($a_2$) are then determined by applying the ratio between daily rainfall or snowfall after quantile mapping ($A_2$) and daily rainfall or snowfall before quantile mapping ($A_1$) to the hourly rainfall or snowfall before quantile mapping ($a_1$):

$$a_2 = a_1 \times \frac{A_2}{A_1} \tag{5}$$

If $A_1 = 0$ and $A_2 = 0$, then $a_2 = 0$. If $A_1 = 0$ and $A_2 \neq 0$, then $a_2 = A_2$.

9. Final adjusted dataset: The resulting adjusted hourly time series for each variable are obtained for each snow year, matching the format of the observational dataset.



## 2.2 SAFRAN reanalysis and application of ADAMONT method using SAFRAN

Although the ADAMONT method is highly generic and can be applied using any hourly-resolution observational dataset, in the following we focus on the use of ADAMONT using the SAFRAN

reanalysis data as an observational dataset. We first describe SAFRAN, then we present specific features of the ADAMONT method when using SAFRAN as the observational dataset.

The SAFRAN system is a regional scale meteorological downscaling and surface analysis system (Durand et al., 1993), which provides hourly data of temperature, precipitation amount and phase, specific humidity, wind speed, and shortwave and longwave radiation for each mountain region (or

"massif") in the French Alps (23 massifs, as illustrated in Fig. 1) but also in the French and Spanish Pyrenees and Corsica. Unlike traditional reanalyses, SAFRAN does not operate on a grid, but on French mountain regions subdivided into different polygons known as massifs. Massifs (Durand et al., 1993, 1999) correspond to regions ranging approximately between 500 and 2,000 km$^2$ for which meteorological conditions are assumed to be spatially homogeneous but vary with altitude.

SAFRAN data are available for elevation bands with a resolution of 300 m. It was used by Durand et al. (2009b) to create a meteorological reanalysis over the French Alps by combining the ERA-40 reanalysis (Uppala et al., 2005) with various meteorological observations including in situ mountain stations, radiosondes and satellite data. It was complemented after the end of the ERA-40 reanalysis (2002) by large-scale meteorological fields from the ARPEGE analysis, so that it now spans the

period from 1959 to 2016, making it one of the longest meteorological reanalyses available in the French mountain regions.

When the ADAMONT method is applied using the SAFRAN reanalysis, only one geographic coordinate is used for each massif, corresponding to the center of the massif (see Fig. 1). However, for each massif several altitude levels are considered, which means that depending on the $N$ fac-

tor considered different RCM grid points may be selected for a given massif and altitude. Also, in order to maximise the consistency between massifs after the adjustment process, the dry/wet analogue day criterion used for the time disaggregation of RCM adjusted variables into hourly variable is computed generally for the entire SAFRAN dataset, here in the 23 French Alps massifs. This means that a day is considered dry when the average of all daily precipitation data is below 1 kg

m$^{-2}$ day$^{-1}$, and wet if it falls above the threshold for all massifs and all altitude levels (from an observational perspective), and for all corresponding adjusted RCM grid points (from an adjusted RCM perspective).

## 2.3 SURFEX/ISBA-Crocus model

Crocus (Brun et al., 1989, 1992; Vionnet et al., 2012) is a detailed snowpack model within the SUR-

FEX externalised surface module (Masson et al., 2013). It enables the computation of the exchanges of energy and mass between the snow surface and the atmosphere (radiative balance, turbulent heat





and moisture fluxes, ...), but also between the snowpack and the ground underneath. Similarly to most land surface models, it requires sub-diurnal (ideally hourly) meteorological forcing data including air temperature, humidity, incoming longwave and shortwave radiation, wind speed, as well as rain and snow precipitation. The one-dimensional multilayer physical snow scheme Crocus is able to simulate the evolution of the snowpack over time, by accounting for several processes occurring in the snowpack, such as thermal diffusion, phase changes, metamorphism, etc. The SAFRAN-Crocus model chain has been operationally used for more than 20 years for avalanche hazard forecasting and extensively evaluated over the alpine domain in particular against snow depth observation stations (Durand et al., 1999, 2009b; Lafaysse et al., 2013). Here we apply the Crocus model using either the SAFRAN reanalysis itself, or adjusted fields from a given RCM using the ADAMONT method, in order to compute and compare snow conditions using either driving data. This is both a proof-of-concept of the applicability of the ADAMONT method to generate data appropriate to driving land surface model, and a mean to assess the intervariable consistency of the ADAMONT output given that Crocus is simultaneously sensitive to all meteorological fields and potentially disturbed by inconsistencies in the forcing dataset.

### 2.4 Method evaluation

To evaluate the ADAMONT method, it was applied to the Météo France ALADIN RCM forced by ERA-Interim over the time period from 1980 to 2010. This RCM was run at 12.5 km resolution and we use the daily time resolution output, which is consistent with typical output from EURO-CORDEX RCMs. This simulation was then adjusted against the SAFRAN reanalysis. The spatial domain (2,200 x 2,200 km, centred on France, see Fig. 1) is deliberately smaller than EURO-CORDEX (5,000 x 5,000 km domain covering all of Europe, Fig. 1) although both are on the same order of magnitude, in order to place more emphasis on the method skills than on the output of the RCM itself, especially in terms of chronology. Indeed, the smaller the domain, the more it is constrained by its driving large-scale model (be it a GCM or a reanalysis) (Alexandru et al., 2007).

Performance indicators described below were computed for temperature and total precipitation, but also for the snow depth, which integrates all the meteorological variables considered in the ADAMONT method. Focus was hereby placed on evaluating the ability of the method to correctly represent integrated outputs computed using SURFEX/ISBA-Crocus from meteorological variables adjusted independently of each other. This is often applied to river discharge for downscaling methods used for hydrological applications (e.g., Lafaysse et al., 2014; Olsson et al., 2015).

The method was applied for all 23 massifs of the French Alps and all elevation bands (Fig. 1), totalling 187 massif/altitude configurations. Performance indicators, described below, were either computed spanning all configurations, or focusing on a given altitude level (1200 m and 2100 m) and/or a subset of massifs (the Vercors massif was taken as an example, and computations were also performed separately for the Northern and Southern Alps, respectively).





We specifically tested the following aspects of the method:

- RCM grid points neighbour selection techniques ($N = 0$ or $N = 50$)

- Learning period: Split-sample evaluation was performed using three different learning and application periods (1980-1995, 1995-2010 and 1980-2010), by evaluating the results on an evaluation period different from the learning period (1995-2010 for simulations with the learning period 1980-1995 and vice-versa). These two sub-periods correspond to markedly different climate conditions in the French Alps (Reid et al., 2015). For simulations using the entire

learning period 1980-2010, the evaluation period was 1980-2010. This case with a 30 years learning period corresponds to the typical duration of the learning period when the method is applied for climate projections.

- Rain/snow quantile mapping: The method was applied with (base case) or without ("no corr") the last adjustment step operating on the rainfall and snowfall separately.

- Raw RCM data: Raw RCM simulations, without any adjustment, were considered for some of the variables (temperature and precipitation only) and compared to adjusted results. This can not be used in the case of snow depth, because daily resolution RCM output cannot be employed to run Crocus.

- The impact of using 6-hour input RCM data instead of daily data was also tested, but yielded

similar results (not shown). Only results based on daily RCM input are presented because GCM/RCM outputs are often available at this time step on data distribution platforms such as the one of EURO-CORDEX

The following indicators were analysed for temperature, total precipitation and snow depth:

- the seasonal average time series from 1980 to 2010 in the SAFRAN and the adjusted RCM

datasets;

- the mean annual cycle over 2 distinct periods: 1980-1995 and 1995-2010 in the SAFRAN and the adjusted RCM datasets;

- the mean value for each elevation band over the evaluation period in the SAFRAN and the adjusted RCM datasets;

- the correlation and the ratio of standard deviations between time series of the SAFRAN and the adjusted RCM datasets for each variable and as a function of the integration window (from 1 day to several years) over the evaluation period;

- the cumulated probability density function (PDF) of daily variables over the evaluation period in the SAFRAN and the adjusted RCM datasets;



– the root mean square error (RMSE) and the mean bias over the evaluation period, computed
over seasonal integration periods based on the SAFRAN and the adjusted RCM datasets;

– scores specific to the detection of precipitation events in the SAFRAN and the adjusted RCM
datasets over the evaluation period: the probability of detection (POD = $n_{hh}/(n_{hh} + n_{hd})$),
the false alarm rate (FAR = $n_{dh}/(n_{dh} + n_{hh})$), the probability of false detection (POFD =
$n_{dh}/(n_{dh} + n_{dd})$) and the true skill score (TSS = POD - FAR), where $n_{hh}$ is the number of
days which are wet in the SAFRAN and wet in the adjusted RCM, $n_{dd}$ the number of days
which are dry in the reanalysis and dry in the adjusted RCM, $n_{hd}$ the number of days which
are wet in the reanalysis but dry in the adjusted RCM and $n_{dh}$ the number of days which are
dry in the reanalysis but wet in the adjusted RCM (a threshold of $1 \, \mathrm{kg \, m^{-2} \, d^{-1}}$ was considered
for the occurrence of precipitation);

– scores for the duration and persistence of precipitation events over the evaluation period
(Wilby et al., 1998; Boe et al., 2006): the relative error on the probability of a dry day (EPD =
$(n_d^R - n_d^S)/n_d^S$), the relative error on the probability of a dry day following a dry day (EPDD =
$(n_{d-d}^R/n_d^R - n_{d-d}^S/n_d^S)/(n_{d-d}^S/n_d^S)$), the relative error on the probability of a wet day follow-
ing a wet day (EPHH = $(n_{h-h}^R/(n-n_d^R) - n_{h-h}^S/(n-n_d^S))/(n_{h-h}^S/(n-n_d^S))$) and the relative
error on the mean duration of wet periods (EHD = $(hdur^R - hdur^S)/hdur^S$), where $n_d^R$ and
$n_d^S$ are the number of dry days in the adjusted RCM and in SAFRAN respectively, $n_{d-d}^R$ and
$n_{d-d}^S$ the number of dry days following a dry day in the adjusted RCM and in SAFRAN re-
spectively, $n_{h-h}^R$ and $n_{h-h}^S$ the number of wet days following a wet day in the adjusted RCM
and in SAFRAN respectively, $n$ is the total number of days, and $hdur^R$ and $hdur^S$ the du-
ration of wet periods in the adjusted RCM and in SAFRAN respectively. A threshold of 1 kg
$\mathrm{m^{-2} \, d^{-1}}$ was considered for the occurrence of precipitation.

These indicators are classically employed (e.g., Boe et al., 2006; Vrac et al., 2007b; Lafaysse,
2011; Kotlarski et al., 2014) to assess:

1. the ability of a model/method to reproduce the statistical characteristics of the observed me-
teorological variables (through the RMSE, the mean bias, the ratio of standard deviations,
the duration and persistence of precipitation events and the cumulated PDFs) and their spa-
tial variability (through the mean values at each elevation band and the analysis of different
massifs);

2. its capacity to reproduce the low frequency variability of the observations, i.e. their chronol-
ogy (through the analysis of seasonal average time series, the correlation as a function of the
integration window, the detection of precipitation events);



3. its temporal transferability, i.e. its ability to reproduce the observed variables over a period
   different from the learning period (through the use of split-sample evaluation, the analysis of
the mean annual cycle over two distinct periods, the seasonal average time series);

4. its inter-variable consistency, which is assessed here by applying the evaluation indicators to
   snow depth, an integrated output of the Crocus model.

When available, we compare the indicators with the same criteria applied to analog resampling
based or transfer function algorithms by Lafaysse (2011) and Lafaysse et al. (2014), and for other
downscaling and adjustment methods by Vrac et al. (2012) and Olsson et al. (2015).

Table 1 outlines the input and output variables of Crocus. Table 2 presents a summary of the
different configurations used for the evaluation.

## 3   Results

### 3.1   Spatial variability and statistical characteristics of the variables

This section provides the evidence needed to assess the performance of the ADAMONT method
applied to a RCM driven by a global reanalysis (ERA-Interim) using the SAFRAN meteorological
reanalysis as the observational dataset in the French Alps. Adjusted RCM data are compared to
SAFRAN itself. Adequate performance of the method is attained when the two datasets match most.

Figure 3 presents the location of the Vercors massif and its average temperature, precipitation and
snow depth for each elevation band, for the evaluation period in the SAFRAN/Crocus reanalysis
as well as adjusted RCM. The shape of the mean altitudinal evolution of all three variables is well
represented compared to SAFRAN, which is also the case for other massifs (see Supplementary
Information). The computed temperature values are very similar to the one in SAFRAN. It is less
the case for precipitation, with over- or underestimation depending on the learning period (Fig. 3)
and the massif considered (Supplementary Information). Despite the differences in the magnitude
of average precipitation in the adjusted RCM compared to SAFRAN, the magnitude of average
snow depth in the different adjusted RCM simulations is remarkably close to the results obtained
using the reanalysis as meteorological input, with slight differences depending on the massif (see
Supplementary Information). For all variables and all massifs, the difference between simulations
using the two RCM grid points neighbour selection techniques ($N = 0$ or $N = 50$) is smaller than
the difference induced by using different learning periods.

Figs. 4-6 display the mean bias and the RMSE for each raw and adjusted RCM simulation com-
pared to SAFRAN, for temperature, precipitation and snow depth, for the Vercors massif. Addition-
ally, Table 3 presents the corresponding scores at the annual time scale compared to mean values, for
the adjusted RCM L. 1980-2010 simulation, for each massif in the French Alps and for the Northern
and Southern Alps, at 1200 m a.s.l. and 2100 m a.s.l.. This highlights the large biases and RMSEs



values obtained when using raw RCM simulations compared to adjusted simulations (Figs. 4-5 and Supplementary Information).

For temperature, biases of the adjusted RCM simulations vary with elevation and for the different
massifs (Fig. 4, Table 3 and Supplementary Information), but lie always within 1 K. Biases are generally smaller in autumn (SON) than for other seasons. RMSEs also vary with elevation and massifs, and can differ significantly between simulations using the two different RCM grid points neighbour selection techniques. For elevations above $\approx 2100$ m a.s.l., stronger biases and higher RMSEs are found for simulations using the selection technique accounting for altitude differences ($N = 50$),
especially in summer (JJA) than for other seasons. Temperature biases and RMSEs values also depend on the learning period considered, the longer learning period 1980-2010 generally presenting smaller biases and RMSEs (Fig. 4 and Supplementary Information).

For precipitation, biases generally vary with altitude (Fig. 5, Table 3 and Supplementary Information), but less than for temperature (Fig. 4, Table 3 and Supplementary Information). Biases of the
adjusted simulations remain smaller than 150 kg m$^{-2}$ per month in absolute value, and are generally stronger in summer. RMSEs values generally increase with altitude. Using different RCM grid points neighbour selection techniques has less impact on precipitation scores than for temperature, except that the $N = 50$ configuration yields more variability in scores with altitude. This is due to the choice of different grid points for different altitudes of a single massif, because precipitation
is spatially more variable than temperature. The influence of the learning period on scores is also visible.

For snow depth, the biases never exceed 50 cm (Fig. 6, Table 3 and Supplementary Information). The biases are smaller in autumn than for other seasons, similar to temperature (Fig. 4 and Supplementary Information). Summer biases at high altitudes are almost always negative, which cannot
always be explained by a combination of positive biases in temperature and/or negative biases in precipitation, indicating the possible impact of other variables on snow depth (such as longwave radiation for example). RMSE values increase with altitude, due to the effect of increased snow accumulation with altitude. Using the $N = 50$ configuration generally degrades scores at high elevations, similar to the effect on temperature.

For precipitation and snow depth, simulations without the ultimate quantile mapping on snowfall and rainfall are also presented (by definition it has no impact on temperature). It is clear from Fig. 5 and the Supplementary Information that without this ultimate correction (no corr), biases in precipitation and snow depth are much stronger and RMSEs much higher than when this correction is applied.

Fig. 7 represents the ratio of standard deviations between each adjusted RCM simulation and SAFRAN for temperature, precipitation and snow depth and as a function of the integration window (from 1 day to several years), over the learning period. Ratios are displayed for the Vercors massif, for altitudes of 1200 m a.s.l. and 2100 m a.s.l.. If this ratio is lower than 1, it means that adjusted





RCM simulations have a smaller standard deviation (i.e. variability) than SAFRAN. For tempera-
ture, the ratio of standard deviations is very close to 1 for integration windows of 1 day to a few
months. It varies more for longer integration windows of 1 year or more. The differences between
the two altitudinal levels considered or between massifs are limited (Fig. 7 and Supplementary Infor-
mation). Similarly, choosing different learning periods or different grid points neighbour selection
techniques has little effect on the ratio of standard deviations. For precipitation, ratios of standard
deviations are also close to 1 (generally between 0.8 and 1.2) for integration windows of 1 day to 1
month. This result is similar to ratios of variance between daily RCMs adjusted with a Cumulative
Distribution Function-transform and observations for the Mediterranean region in Vrac et al. (2012).
For integration windows of 1 month or more, the ratios vary more, with under- or overestimation
of variance depending on the massif, the learning period and the grid points neighbour selection
technique considered (Fig. 7 and Supplementary Information). For snow depth, the ratio does not
vary until 1 month of integration approximately (Fig. 7 and Supplementary Information), and shows
larger variations for higher values. Some differences can be noted for different altitudes, and dif-
ferent massifs, but also for different learning periods and the two grid points neighbour selection
techniques considered.

Fig. 8 presents the cumulated probability density functions (PDFs) of daily temperature, precipi-
tation and snow depth at 1200 m a.s.l. and 2100 m a.s.l. for the Vercors massif. The distributions of
daily temperature of adjusted RCM simulations are remarkably close to the distribution of SAFRAN
(Fig. 8 and Supplementary Information). The agreement is better than the one observed in Lafaysse
(2011) and Lafaysse et al. (2014) between the different configurations of analog-based and trans-
fer functions algorithms and SAFRAN for the Durance basin (see Fig.F.2 in Lafaysse (2011), and
Fig.5 in Lafaysse et al. (2014)). A similar agreement was observed in Olsson et al. (2015) between
two configurations of a distribution-based scaling method and observations in Finland. Only small
differences are observed for different altitudes or different massifs (Fig. 8 and Supplementary Infor-
mation), and the choice of the learning period or the grid points neighbour selection technique has
almost no impact on the PDF. For precipitation, the PDFs of adjusted RCM simulations are also very
close to the PDF of SAFRAN, with a slight overestimation or underestimation of moderate to high
precipitation, depending on the learning period, occurring for most massifs (Fig. 8 and Supplemen-
tary Information). This result is similar to that observed in Lafaysse (2011) for the Durance basin
(see Fig.11.7 therein). As for temperature, altitude and massif location have only a small impact on
the distribution, as well as the grid points neighbour selection technique considered. The distribution
of snow depth, on the other hand, depends more on the massif considered and the altitude (Fig. 8 and
Supplementary Information). As for precipitation, the moderate to high snow depth values seem to
be slightly overestimated or underestimated for most massifs, depending on the learning period. The
choice of the grid points neighbour selection technique has also slightly more impact on snow depth
PDFs than for temperature and precipitation. The fact that PDFs for temperature and precipitation are





very close to the ones of SAFRAN is a logical consequence of using a quantile mapping approach. That it is also true for snow depth indicates that even if they are treated separately, the inter-variable consistency of the meteorological fields generated using our method is in general appropriate.

The capacity to reproduce the duration and persistence of precipitation events is shown in Fig. 9.
The ratio between the number of dry days and the number of rainy or snowy days is very correctly reproduced for every massif and altitude (Fig. 9 and Supplementary Information), the relative error on the probability of a dry day being lower than 5%. This feature was also observed by Lafaysse (2011) in his study of the Durance basin (see Fig.11.10 therein). The persistence of dry and rainy/snowy events is generally underestimated (up to about -30%), which was also the case in Lafaysse (2011),
even though the error depends on the massif and the altitude considered. In general, errors on the persistence of precipitation events are larger in massifs of the Southern Alps than the Northern Alps (Supplementary Information). Using different learning periods and different grid points neighbour selection techniques has an impact on scores, but this is small compared to the influence of the massif or the altitude.

**3.2 Mean seasonal variations**

Fig. 10 represents the mean annual cycle of temperature, precipitation and snow depth for the different adjusted RCM simulations vs. the SAFRAN/Crocus reanalysis, for the period 1980-1995 and 1995-2010, for the Vercors massif at 1200 m a.s.l. and 2100 m a.s.l.. The mean annual cycle of temperature is very well reproduced for every massif and altitude (Fig. 10 and Supplementary In-
formation). Using different grid points neighbour selection techniques has a limited impact on the mean annual cycle. For precipitation, the mean annual cycle is relatively well reproduced (Fig. 10 and Supplementary Information). The choice of grid points neighbour selection technique can have slightly more influence on the results than for temperature. For snow depth, the annual cycle is remarkably well reproduced, with peak snow depth in the core of winter (JFM), and no snow or
reduced amounts in late summer months (JAS) (Fig. 10 and Supplementary Information). As for temperature, the impact of the grid points neighbour selection technique is very limited.

**3.3 Interannual variability**

The chronology of time series of seasonal averages of temperature, precipitation and snow depth from 1980 to 2010 is shown in Figs. 11-13, for the Vercors massif at 1200 m a.s.l. and 2100 m a.s.l.,
in SAFRAN and the adjusted RCM. Temperature RCM time series are similar to SAFRAN, with an interannual variability which is well reproduced (Fig. 11 and Supplementary Information). Some significant differences appear when using different learning periods, as already noted in Sect. 3.2. Using different grid points neighbour selection techniques has an impact on the time series of temperature which is generally smaller than the influence of the learning period. However, as already
noted in Sect. 3.1, the agreement between observed and simulated time series is degraded for high





altitudes under the spatial and altitudinal ($N = 50$) grid points neighbour selection technique. The interannual variability of precipitation is also well reproduced for most massifs and altitudes (Fig. 12 and Supplementary Information), especially given that the only forcing of the RCM comes from ERA-Interim reanalysis at the boundaries of the RCM domain. It is slightly less well reproduced

in summer (JJA), as observed by Lafaysse (2011) for the analog resampling based transfer function algorithm DSCLIM (Pagé et al., 2009) and the Durance basin (see Fig.10.1 therein). Differences between simulations using different learning periods mostly appear in summer (JJA). The use of different grid points neighbour selection techniques has a rather limited impact on time series of precipitation, whose magnitude depends on the massif and the altitude (Fig. 12 and Supplementary

Information). For snow depth, the interannual variability is well reproduced in winter (DJF) and correctly reproduced in intermediate seasons (MAM and SON). Summer snow depths are generally underestimated, as already noted in Sect. 3.1, but represent a small portion of the annual snow accumulation. Likewise, adjusted data using the spatial and altitudinal ($N = 50$) RCM grid points selection technique can be degraded at high altitudes, similarly to temperature.

Fig. 14 displays the temporal correlation between each adjusted RCM simulation and SAFRAN over the evaluation period for temperature and precipitation and as a function of the integration window (from 1 day to several years). Correlations are displayed for the Vercors massif, for altitudes of 1200 m a.s.l. and 2100 m a.s.l.. Additionally, Table 3 presents the same correlation values at the same altitudes, for an integration window of 1 year, and for the adjusted RCM L. 1980-2010

simulation only, for every massif of the French Alps, and for the Northern and Southern Alps. Snow depth values were not included because of their cumulative nature. Correlations for temperature are very high (always above 0.8) for all massifs and altitudes until an integration window of a few months to 1 year (Fig. 14, Table 3 and Supplementary Information), similar to Lafaysse (2011) (see Fig.F.21 therein). The differences between learning periods are negligible. As already observed in Sect. 3.1

and for the time series above, the correlation is clearly degraded for high altitudes (above ≈2100 m a.s.l.) in simulations using the $N = 50$ grid points selection technique. Precipitation also yields satisfactory correlation values (always above 0.4) until a few months integration window, which vary depending on the massif considered(Fig. 14 and Supplementary Information). Correlations are generally similar or even better than the ones observed in Lafaysse (2011) for various statistical

downscaling models and different configurations of the ALADIN RCM (see Fig.12.10 therein). The use of the $N = 50$ grid points neighbour selection technique increases or decreases correlation values depending on the massif and the altitude considered. The choice of learning period has a limited effect on correlation, at least up to integration windows of a few months. Correlations are higher at the scale of the Northern and Southern Alps than at the massif scale (Table 3). This scale dependence

of precipitation downscaling skill was also illustrated by Gangopadhyay et al. (2004) and Mezghani and Hingray (2009).




Scores for the detection of precipitation events are presented in Fig. 15, for the Vercors massif, for altitudes of 1200 m a.s.l. and 2100 m a.s.l.. The scores vary depending on massifs and altitude, but a general pattern emerges (Fig. 15 and Supplementary Information). The POD is the highest, with values between 0.55 and 0.8, very similar to Lafaysse (2011) (see Figs. 11.14 and 12.8 therein). The FAR is rather low (always below 0.5), as well as the POFD, below 0.2. TSS are generally better for massifs of the Northern Alps than the Southern Alps (Supplementary Information), where PODs are lower and FAR much higher. Such results indicate that the method performs well in detecting precipitation events. Using different learning periods has a rather limited impact on the detection of precipitation. The choice of the grid points selection technique has a limited influence at low to mid-altitudes, which increases above ≈2100 m a.s.l..

## 4  Discussion

This section discusses the main limits of the method described and evaluated here, and the limits of the evaluation method itself.

### 4.1  Transferability in time

The temporal transferability of the ADAMONT method, i.e. its capacity to apply adequately to a period which is different from the learning period, can be evaluated from results in Sects. 3.1, 3.3 and 3.2.

Figs. 11-13, 10 and Supplementary Information reveal some significant differences when using different learning periods. This feature is generally most visible in summer. It denotes a limit in the temporal transferability of the ADAMONT method, which was also the case in Lafaysse (2011) for the analog-based and transfer functions algorithms (see Figs. 11.11 and 11.12 therein). Using the longer learning period 1980-2010 yields better results, most probably due to the fact that, in this case, the learning and evaluation periods are the same, but also the fact that the learning period is longer.

There are some limits in the conclusions which can be drawn from this transferability assessment. First, reanalysis data used here as forcing for the RCM (ERA-Interim) or for statistical adjustment and evaluation purposes (SAFRAN reanalysis) are heterogeneous in time (Sterl, 2004; Vidal et al., 2010). These heterogeneities are especially marked in summer in the SAFRAN reanalysis, when most observations from mountain stations are not available. Secondly, variations which will occur in the future climate are expected to be much stronger than the variations which can be tested on our evaluation period. Issues related to the time transferability of the adjustment approach may be amplified when applied in the context of climate projections, but their relative impact will probably be lower than shown here given the magnitude of the expected changes.





### 4.2  Impact of the spatial selection technique

The impact of the RCM grid point selection technique is illustrated in Sects. 3.1 and 3.3. Indeed, Figs. 4-6, 11-13, 14 and Supplementary Information show a clear degradation of scores for elevations above $\approx 2100$ m a.s.l. using a selection criterion explicitly accounting for the altitude difference ($N = 50$). This is linked to the scarcity of high altitude grid points in ALADIN compared to SAFRAN, resulting in grid points being selected several tens of kilometres from the centre point of most SAFRAN massifs (see Fig. 3 and Supplementary Information for the location of selected grid points). The impact of this issue depends on the location of massifs relative to high-altitude grid points in ALADIN. For example, most Southern Alps massifs are affected, except the southernmost massifs of Ubaye, Alpes Azur and Mercantour (Supplementary Information), which are located less than 15 km from high-altitude points. This shows that, although it seems appealing to select RCM grid points at elevations matching the elevation of the observation dataset, rather than using RCM grid points with a potentially large elevation difference (hence leading to stronger adjustment requirement), in practice the results are far more homogeneous and quantitatively generally equivalent or better when concentrating only on the horizontal distance between the RCM grid points and the observation dataset.

### 4.3  Inter-variable consistency

The lack of explicitly enforced inter-variable consistency of the quantile mapping method can be a major disadvantage. As we focus on a mountainous region for the evaluation and future use of the method the consistency between temperature and precipitation phase is crucial. The impact of this ultimate correction is assessed in Sect. 3.1. Figs. 5-6 and Supplementary Information show that without this ultimate correction (no corr), biases are much stronger and RMSEs much higher than with this ultimate correction, highlighting its importance.

The inter-variable consistency of the ADAMONT method is indirectly assessed by applying the evaluation metrics described above to an integrated output of the Crocus model, the snow depth, which is computed from meteorological variables adjusted independently from each other. As mentioned above, snow depth results are generally satisfying, which tend to indicate a good inter-variable consistency. Performance indicators for snow depths are often consistent with temperature and precipitation indicators, even though they cannot always be explained by these two variables alone (for example the analysis of biases in Sect. 3.1), indicating the probable influence of other variables not directly analysed here such as longwave radiation.

### 4.4  Limits of the evaluation method

The spatial consistency of the ADAMONT method has not been evaluated other than by using spatial averages. In future studies, it would be necessary to test it by evaluating spatial correlations (for





example using metrics described in Kotlarski et al. (2014)), or by using integrated variables requiring
spatial variability, such as snow cover area or river discharges.

In this study, we evaluated the method using only the ALADIN-Climate RCM. However, Olsson
et al. (2015) showed that the choice of RCM could have a significant impact on the evaluation of the
performance of the adjustment method. Evaluation using another RCM could thus prove useful, even
though we would have to use RCM outputs run on the same spatial domain as the ALADIN-Climate
RCM in order to compare them.

## 5   Conclusions

The new method to statistically adjust regional climate model projections ADAMONT is introduced,
which provides hourly adjusted outputs of temperature, precipitation, wind speed, humidity and
short- and longwave radiation necessary to force energy balance land surface (impact) models. The
method processes daily outputs from an RCM and adjusts them against a sub-daily (hourly, typi-
cally) observational dataset. The method was evaluated using outputs from the ALADIN-Climate
RCM driven by ERA-Interim reanalysis for the time period 1980 - 2010, using the SAFRAN mete-
orological reanalysis in the French Alps as an observation dataset. The direct outputs of the ADA-
MONT method, namely temperature and total precipitation, as well as an indirect output, namely
snow depth, computed by the Crocus model from meteorological variables corrected independently
of each other, were evaluated. The impact of the learning period was tested, as well as the method
to select RCM grid points corresponding to each observational point. The evaluation addressed four
main concerns: (1) the ability of the ADAMONT method to reproduce the spatial (especially altitu-
dinal) variability and the statistical characteristics of SAFRAN variables, (2) its ability to reproduce
the low frequency variability, i.e. the chronology, of SAFRAN, through the analysis of the inter-
annual variability and the annual cycle of adjusted variables (3) the temporal transferability of the
method, and (4) its inter-variable consistency.

Performance scores are always better for adjusted RCM simulations than for raw RCM simula-
tions, which highlights the need for such adjustment and demonstrates the skill of the method. In
general, the performance of the ADAMONT method concerning temperature is better than for pre-
cipitation. However, evaluation indicators for precipitation are generally similar or even better than
the indicators evaluated in Lafaysse (2011) and Lafaysse et al. (2014) for other types of algorithms
(analog-based or transfer functions). Snow depth yields good results, considering its integrated na-
ture, i.e. the fact that it was computed from variables corrected independently. The impact of the
learning period depends on the evaluation indicator considered, and must be considered when ap-
plying the method. The best solution is probably to choose the longest possible learning period. For
precipitation and snow depth, the importance of the ultimate quantile mapping applied to snowfall
and rainfall is unambiguously demonstrated. Using a grid points selection technique relying on spa-





tial but also altitudinal proximity between SAFRAN massif centre points and RCM grid points either
had no impact on the performance indicators or degraded them for altitudes higher than 2100 m a.s.l..
As a consequence, the simple spatial grid points neighbour selection technique will be retained for
future applications of the method.

The ADAMONT method is generic and can be applied to any observational dataset. Its application
using the SAFRAN reanalysis as the observation dataset is somewhat a specific case, initially tailord
for French mountainous regions (Durand et al., 2009a). However, beyond the French mountain re-
gions, the method could be applied in France using the SAFRAN-France gridded reanalysis (Vidal
et al., 2010). A Spanish version of SAFRAN was also developed recently (Quintana-Seguí et al.,
2017).The method could be applied to other observational datasets or meteorological reanalyses,
such as ERA-Interim surface fields (Dee et al., 2011) or MESCAN (Soci et al., 2016).

## 6    Code availability

The code of the ADAMONT v1.0 method is available as an open git repository after free
registration at https://opensource.cnrm-game-meteo.fr/projects/adamont. The version used for
this article is available at https://opensource.cnrm-game-meteo.fr/projects/adamont/repository?rev=
ADAMONT-v1.0.
The version of the open source code of SURFEX/ISBA-Crocus used in this study is avail-
able as a specific branch of an open svn repository, after free registration, at https://opensource.
cnrm-game-meteo.fr/projects/surfex. For reproductibility of results, the version used in this work is
tagged as http://svn.cnrm-game-meteo.fr/projets/surfex/tags/ADAMONT-1.0.

*Acknowledgements.* This study benefited from funding from the French Ministry for Ecology (MEEM) through
the GICC program and ONERC, in the framework of the ADAMONT project. It also forms part of the Interreg
project POCTEFA/Clim'Py. CNRM/CEN is part of LabEX OSUG@2020 (ANR10 LABX56).




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



**Table 1.** Variables considered in this study : Variable name, Abbreviation, Input or Output of Crocus, Units, Level and Method of integration (of the observational dataset from hourly to daily) and disaggregation (RCM adjusted data from daily to hourly). Variables used for the evaluation of the ADAMONT method are highlighted in bold characters. SW = shortwave, LW = longwave.

| Variable | Abbreviation | Input/Output | Units | Level | Method |
|---|---|---|---|---|---|
| **Temperature** | Tair | Input | K | 2 m | Min, max |
| Specific Humidity | Qair | Input | $kg\,kg^{-1}$ | 2 m | Last value |
| Wind speed | Wind | Input | $m\,s^{-1}$ | 10 m | Last value |
| **Rainfall Rate** | Rainf | Input | $kg\,m^{-2}\,h^{-1}$ | Surface | Mean |
| **Snowfall Rate** | Snowf | Input | $kg\,m^{-2}\,h^{-1}$ | Surface | Mean |
| Incident LW Radiation | LWdown | Input | $W\,m^{-2}$ | Surface | Mean |
| Incident Direct SW Radiation | DIR_SWdown | Input | $W\,m^{-2}$ | Surface | Mean |
| Incident Diffuse SW Radiation | SCA_SWdown | Input | $W\,m^{-2}$ | Surface | Mean |
| **Snowpack Depth** | SNOWDEPTH | Output | m | < Surface | - |





**Table 2.** Name and description of the different configurations used in the evaluation of the ADAMONT method.

| Name | Description |
| --- | --- |
| SAFRAN reanalysis | Simulation carried out with the SAFRAN reanalysis, over the period considered in the figures (1980-2010, 1980-1995 or 1995-2010) |
| RCM raw | Simulation carried out over the period considered in the figures, with the raw ALADIN RCM (without adjustment) |
| RCM raw N50 | Simulation carried out over the period considered in the figures, with the raw ALADIN RCM (without adjustment), using the spatial and altitudinal RCM grid points neighbour selection technique ($N = 50$) |
| RCM L. 1980-1995 | Simulation carried out over the period considered in the figures, with the ALADIN RCM, and the learning period 1980-1995 |
| RCM L. 1980-1995 N50 | Simulation carried out over the period considered in the figures, with the ALADIN RCM, and the learning period 1980-1995, using the spatial and altitudinal RCM grid points neighbour selection technique ($N = 50$) |
| RCM L. 1980-1995 no corr | Same as RCM L. 1980-1995, but without performing the last quantile mapping for rain and snow |
| RCM L. 1995-2010 | Simulation carried out over the period considered in the figures, with the ALADIN RCM, and the learning period 1995-2010 |
| RCM L. 1995-2010 N50 | Simulation carried out over the period considered in the figures, with the ALADIN RCM, and the learning period 1995-2010, using the spatial and altitudinal RCM grid points neighbour selection technique ($N = 50$) |
| RCM L. 1995-2010 no corr | Same as RCM L. 1995-2010, but without performing the last quantile mapping for rain and snow |
| RCM L. 1980-2010 | Simulation carried out over the period considered in the figures, with the ALADIN RCM, and the learning period 1980-2010 |
| RCM L. 1980-2010 N50 | Simulation carried out over the period considered in the figures, with the ALADIN RCM, and the learning period 1980-2010, using the spatial and altitudinal RCM grid points neighbour selection technique ($N = 50$) |
| RCM L. 1980-2010 no corr | Same as RCM L. 1980-2010, but without performing the last quantile mapping for rain and snow |



**Table 3.** Mean values and scores of the ADAMONT-adjusted RCM L. 1980-2010 simulation compared to
SAFRAN over the period 1980-2010 for each massif of the French Alps and for the Northern and Southern
Alps, at 1200 m and 2100 m elevation : mean annual temperature (T, in K) and precipitation (P, in kg m$^{-2}$
yr$^{-1}$), mean winter (DJFMAM) snow depth (SD, in m), mean annual bias of T and P, mean winter bias of SD,
annual root mean square error (RMSE) of T and P, winter RMSE of SD, and annual correlation of T and P.

| Massif | Altitude | Mean value | | | Mean bias | | | RMSE | | | Correlation | |
|---|---|---|---|---|---|---|---|---|---|---|---|---|
| | | T | P | SD | T | P | SD | T | P | SD | T | P |
| Northern Alps | 1200 m | 280.1 | 991 | 0.32 | -0.04 | -217 | -0.04 | 0.40 | 643 | 0.11 | 0.99 | 0.92 |
| | 2100 m | 275.8 | 675 | 1.25 | 0.03 | -294 | -0.03 | 0.50 | 804 | 0.16 | 0.96 | 0.91 |
| Chablais | 1200 m | 279.5 | 1247 | 0.40 | -0.05 | -233 | -0.04 | 0.56 | 1010 | 0.16 | 0.97 | 0.56 |
| | 2100 m | 275.5 | 845 | 1.54 | 0.07 | -313 | -0.04 | 0.55 | 1222 | 0.27 | 0.95 | 0.52 |
| Aravis | 1200 m | 279.8 | 1205 | 0.42 | -0.02 | -282 | -0.05 | 0.47 | 1021 | 0.17 | 0.98 | 0.88 |
| | 2100 m | 275.7 | 814 | 1.65 | 0.07 | -389 | -0.03 | 0.56 | 1310 | 0.29 | 0.95 | 0.88 |
| Mont Blanc | 1200 m | 279.7 | 1104 | 0.35 | -0.06 | -232 | -0.04 | 0.55 | 981 | 0.12 | 0.97 | 0.58 |
| | 2100 m | 275.6 | 854 | 1.44 | 0.04 | -367 | -0.10 | 0.51 | 1316 | 0.29 | 0.97 | 0.59 |
| Bauges | 1200 m | 279.7 | 1177 | 0.44 | -0.02 | -273 | -0.04 | 0.44 | 948 | 0.17 | 0.98 | 0.90 |
| | 2100 m | 275.6 | 751 | 1.65 | 0.07 | -408 | 0.01 | 0.56 | 1099 | 0.31 | 0.95 | 0.90 |
| Beaufortin | 1200 m | 280.1 | 921 | 0.40 | -0.02 | -195 | -0.02 | 0.45 | 786 | 0.14 | 0.98 | 0.79 |
| | 2100 m | 275.6 | 653 | 1.36 | 0.05 | -291 | -0.05 | 0.53 | 974 | 0.20 | 0.96 | 0.78 |
| Haute Tarentaise | 1200 m | 280.3 | 727 | 0.33 | -0.04 | -177 | -0.05 | 0.65 | 686 | 0.16 | 0.97 | 0.75 |
| | 2100 m | 275.4 | 509 | 1.01 | 0.00 | -199 | -0.06 | 0.62 | 789 | 0.25 | 0.97 | 0.74 |
| Chartreuse | 1200 m | 280.0 | 1225 | 0.37 | -0.02 | -303 | -0.04 | 0.51 | 1070 | 0.21 | 0.97 | 0.87 |
| | 2100 m | 276.1 | 761 | 1.57 | 0.07 | -409 | 0.06 | 0.75 | 1307 | 0.30 | 0.89 | 0.84 |
| Belledonne | 1200 m | 280.1 | 1112 | 0.34 | -0.05 | -229 | -0.03 | 0.48 | 917 | 0.16 | 0.98 | 0.89 |
| | 2100 m | 275.9 | 771 | 1.45 | 0.03 | -314 | 0.05 | 0.66 | 1175 | 0.26 | 0.91 | 0.88 |
| Maurienne | 1200 m | 280.4 | 854 | 0.33 | -0.04 | -184 | -0.01 | 0.48 | 767 | 0.15 | 0.99 | 0.84 |
| | 2100 m | 275.8 | 548 | 1.10 | 0.03 | -241 | -0.02 | 0.55 | 868 | 0.21 | 0.95 | 0.85 |
| Vanoise | 1200 m | 280.4 | 771 | 0.31 | -0.03 | -129 | -0.02 | 0.53 | 694 | 0.11 | 0.98 | 0.82 |
| | 2100 m | 275.6 | 549 | 1.00 | 0.00 | -186 | -0.04 | 0.54 | 833 | 0.20 | 0.96 | 0.81 |
| Haute Maurienne | 1200 m | 280.7 | 642 | 0.15 | -0.05 | -147 | -0.04 | 0.59 | 693 | 0.10 | 0.97 | 0.87 |
| | 2100 m | 275.5 | 487 | 0.61 | -0.03 | -185 | -0.08 | 0.48 | 858 | 0.19 | 0.98 | 0.84 |
| Grandes Rousses | 1200 m | 280.4 | 907 | 0.26 | -0.06 | -200 | -0.03 | 0.65 | 902 | 0.14 | 0.96 | 0.83 |
| | 2100 m | 276.0 | 591 | 1.06 | 0.00 | -244 | -0.02 | 0.76 | 998 | 0.26 | 0.88 | 0.85 |
| Vercors | 1200 m | 280.2 | 1032 | 0.20 | -0.02 | -228 | -0.04 | 0.50 | 768 | 0.13 | 0.97 | 0.89 |
| | 2100 m | 276.2 | 686 | 1.21 | 0.05 | -308 | -0.03 | 0.73 | 971 | 0.25 | 0.88 | 0.85 |



| Massif | Altitude | Mean value | | | Mean bias | | | RMSE | | | Correlation | |
|---|---|---|---|---|---|---|---|---|---|---|---|---|
| | | T | P | SD | T | P | SD | T | P | SD | T | P |
| Oisans | 1200 m | 280.5 | 947 | 0.19 | -0.03 | -223 | -0.05 | 0.49 | 903 | 0.11 | 0.98 | 0.84 |
| | 2100 m | 276.2 | 629 | 0.91 | 0.01 | -264 | -0.07 | 0.65 | 1038 | 0.28 | 0.93 | 0.85 |
| Southern Alps | 1200 m | 281.2 | 775 | 0.10 | 0.03 | -150 | -0.02 | 0.49 | 530 | 0.05 | 0.98 | 0.93 |
| | 2100 m | 276.4 | 546 | 0.63 | 0.02 | -194 | -0.04 | 0.47 | 646 | 0.15 | 0.98 | 0.93 |
| Thabor | 2100 m | 275.9 | 452 | 0.70 | 0.00 | -220 | -0.03 | 0.61 | 868 | 0.20 | 0.96 | 0.87 |
| Pelvoux | 1200 m | 280.9 | 733 | 0.20 | 0.00 | -146 | -0.01 | 0.75 | 676 | 0.08 | 0.94 | 0.93 |
| | 2100 m | 276.2 | 533 | 0.92 | 0.02 | -204 | 0.00 | 0.67 | 878 | 0.24 | 0.94 | 0.92 |
| Queyras | 1200 m | 281.1 | 568 | 0.10 | 0.01 | -138 | -0.03 | 0.56 | 641 | 0.07 | 0.98 | 0.83 |
| | 2100 m | 276.0 | 426 | 0.46 | 0.02 | -163 | -0.04 | 0.54 | 770 | 0.18 | 0.98 | 0.82 |
| Dévoluy | 1200 m | 280.6 | 935 | 0.10 | -0.02 | -171 | -0.03 | 0.50 | 784 | 0.08 | 0.97 | 0.86 |
| | 2100 m | 276.4 | 633 | 0.77 | 0.05 | -186 | -0.02 | 0.66 | 919 | 0.24 | 0.94 | 0.84 |
| Champsaur | 1200 m | 280.8 | 823 | 0.13 | -0.02 | -180 | -0.02 | 0.57 | 705 | 0.08 | 0.98 | 0.90 |
| | 2100 m | 276.4 | 580 | 0.74 | 0.01 | -217 | -0.04 | 0.57 | 880 | 0.24 | 0.97 | 0.88 |
| Parpaillon | 1200 m | 281.1 | 629 | 0.13 | 0.02 | -145 | -0.02 | 0.60 | 644 | 0.07 | 0.97 | 0.87 |
| | 2100 m | 276.4 | 467 | 0.54 | 0.02 | -179 | -0.03 | 0.52 | 736 | 0.17 | 0.99 | 0.87 |
| Ubaye | 1200 m | 281.2 | 682 | 0.06 | 0.04 | -132 | -0.01 | 0.82 | 580 | 0.05 | 0.92 | 0.89 |
| | 2100 m | 276.6 | 525 | 0.43 | 0.03 | -179 | -0.06 | 0.58 | 705 | 0.18 | 0.98 | 0.89 |
| Alpes Azur | 1200 m | 281.7 | 877 | 0.05 | 0.10 | -119 | -0.02 | 0.66 | 728 | 0.08 | 0.94 | 0.78 |
| | 2100 m | 277.0 | 590 | 0.53 | 0.03 | -180 | -0.10 | 0.48 | 854 | 0.22 | 0.98 | 0.78 |
| Mercantour | 1200 m | 282.3 | 952 | 0.05 | 0.09 | -168 | -0.03 | 0.71 | 974 | 0.07 | 0.94 | 0.68 |
| | 2100 m | 276.9 | 707 | 0.56 | -0.02 | -223 | -0.06 | 0.61 | 1133 | 0.27 | 0.96 | 0.69 |



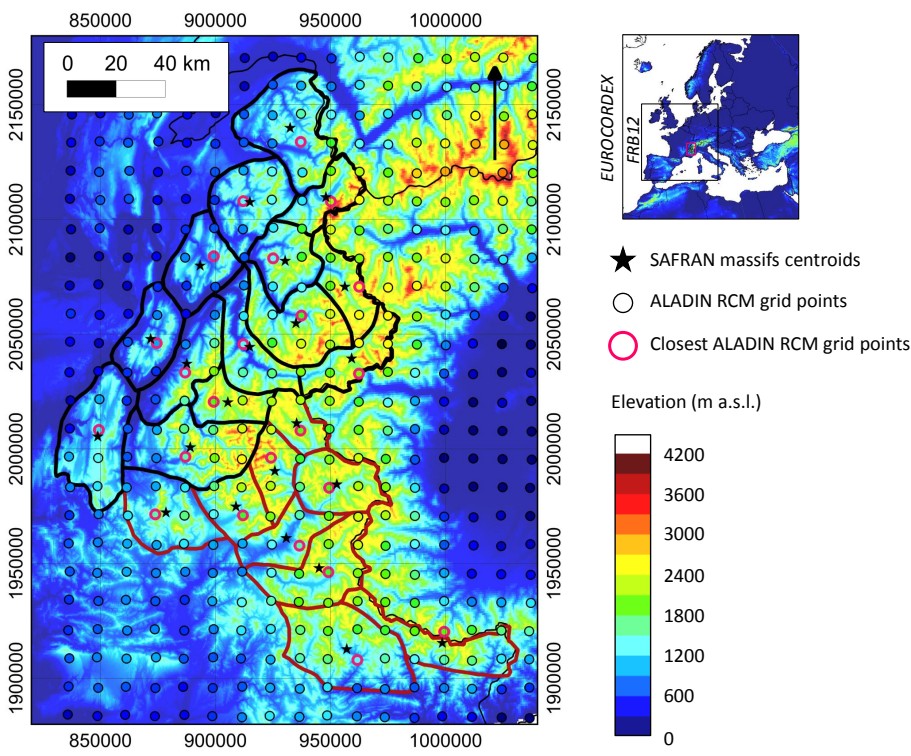

**Figure 1.** Description of geographical configuration of the SAFRAN reanalysis and the ALADIN RCM. The top right panel illustrates the spatial domains covered by the simulation (FRB12) and by EURO-CORDEX, and the location of the study area is indicated by the pink box. In the main panel, SAFRAN massifs are delimited by the black contours for the Northern Alps and by the burgundy contours for the Southern Alps, and their centre points are indicated by the black stars. ALADIN grid points are represented by dots, with pink contour for the grid points closest to each SAFRAN massif centre point. Surface elevation in France is from the 50m-DEM from the Institut National d'Information Géographique et Forestière (IGN) and outside France from GTOPO30 (resolution of 30 arc seconds ≈ 1 km). The elevation of ALADIN grid points is indicated by the color palette (in m above sea level (a.s.l.)). Projection is in Lambert II étendu (L2E).




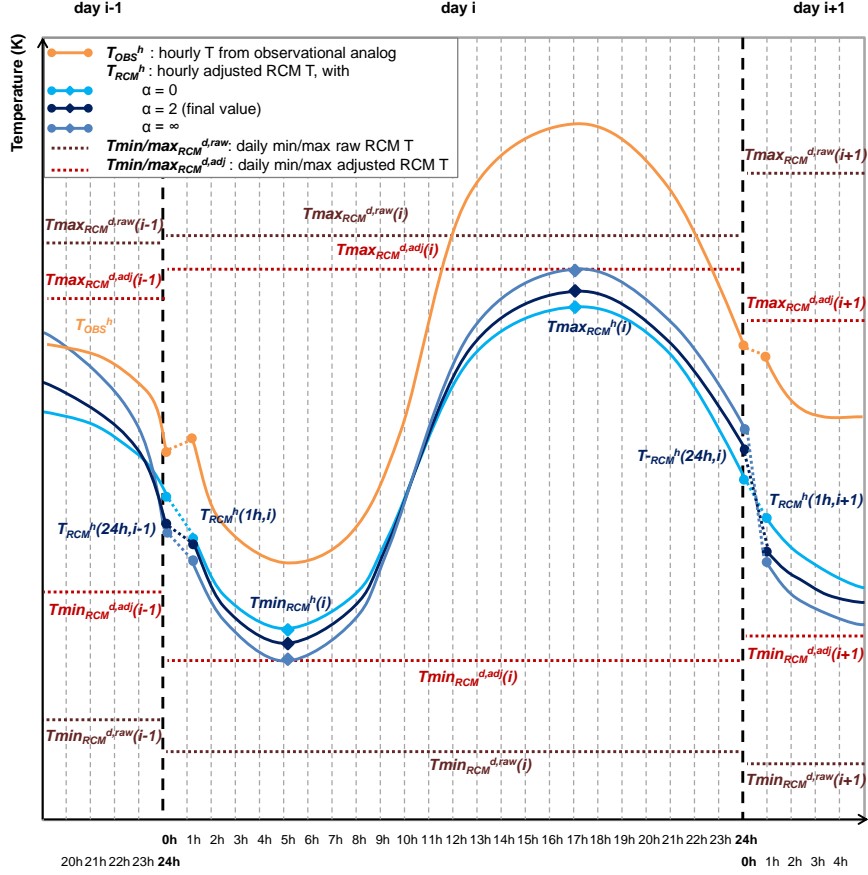

**Figure 2.** Illustration of the different parameters taken into account in the disaggregation of RCM temperature from a daily integration period into an hourly time step. $T^h_{RCM}(1h, i)$ and $T^h_{RCM}(24h, i-1)$ are the hourly adjusted RCM temperature values at the first time step of day i and at the last time step of the day before (i-1), $Tmin^h_{RCM}(i)$ and $Tmax^h_{RCM}(i)$ are the hourly minimum and maximum adjusted RCM temperature values respectively, and $Tmin^{d,adj}_{RCM}(i)$ and $Tmax^{d,adj}_{RCM}(i)$ are the daily minimum and maximum adjusted RCM temperature values respectively. $\alpha$ is a parameter which can be tuned to give more importance to the minimisation of differences between daily and hourly RCM minima and maxima. Hourly adjusted RCM temperature time series for values of $\alpha$ of zero, 2 and infinite are shown. $T^h_{OBS}$ corresponds to the hourly series of the chosen daily analogue, and $Tmin^{d,raw}_{RCM}(i)$ and $Tmax^{d,raw}_{RCM}(i)$ are the daily raw minimum and maximum RCM temperature values (before adjustment).





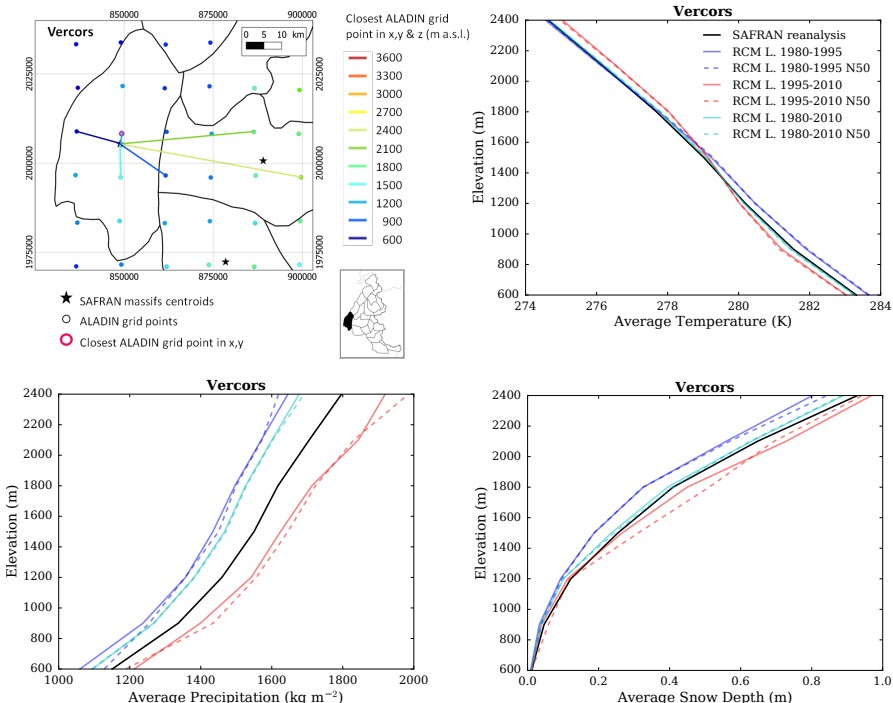

**Figure 3.** (top left) Location of the Vercors massif, with ALADIN RCM grid points chosen as the closest in x, y ($N = 0$, pink contour) and in x, y and z (using $N \neq 0$). Coloured lines link each SAFRAN massif centre point with the corresponding grid point in ALADIN for the different elevations considered (in m above sea level (a.s.l.)). (top right) Mean temperature for each elevation band over the evaluation period in each adjusted RCM simulation (different learning periods and 2 grid points neighbour selection methods) and in SAFRAN. (bottom left) Mean precipitation for each elevation band over the evaluation period. (bottom right) Mean snow depth (using Crocus in this case) for each elevation band over the evaluation period.





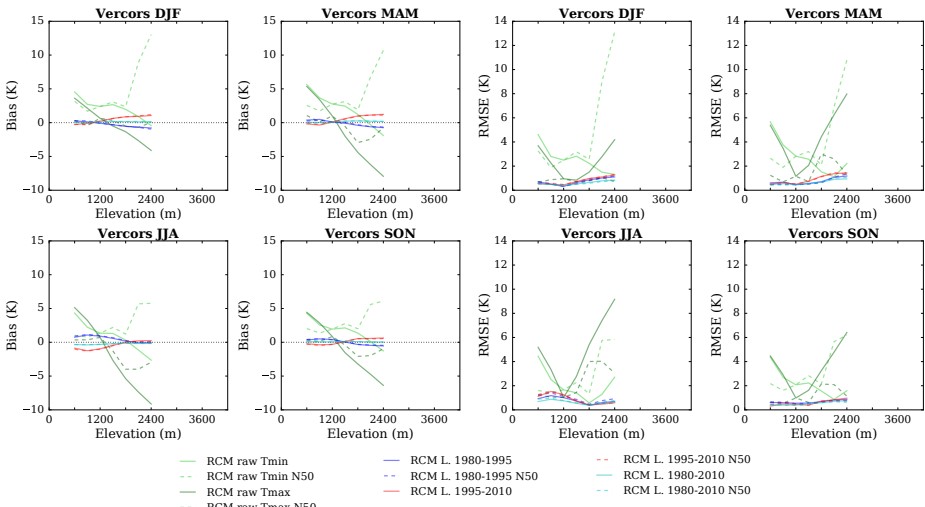

**Figure 4.** Temperature mean bias and root mean square error (RMSE) of each raw and adjusted RCM simulation compared to the SAFRAN reanalysis over the evaluation period for the Vercors massif as a function of elevation. Scores computed for the raw RCM simulations concern minimum and maximum daily temperatures.

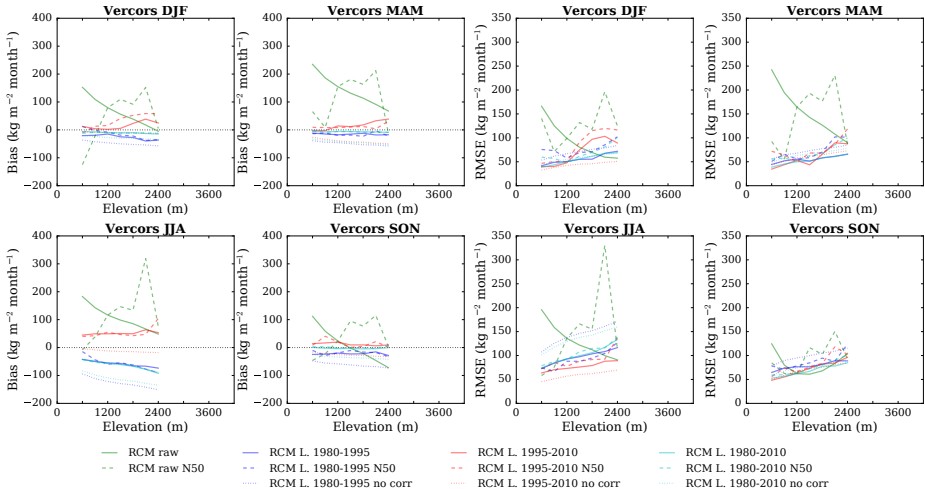

**Figure 5.** Precipitation mean bias and root mean square error (RMSE) of each raw and adjusted RCM simulation compared to the SAFRAN reanalysis over the evaluation period for the Vercors massif as a function of elevation.





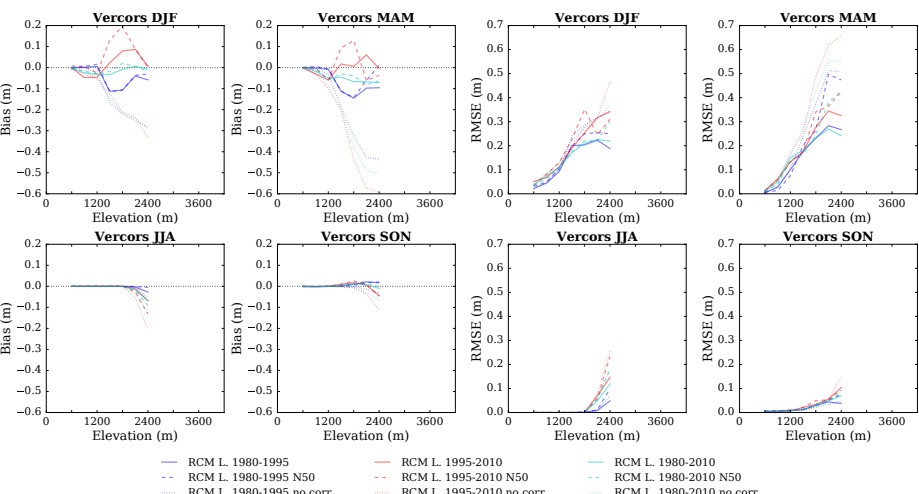

**Figure 6.** Snow depth mean bias and root mean square error (RMSE) of each adjusted RCM simulation (used as input of Crocus) compared to the SAFRAN/Crocus reanalysis over the evaluation period for the Vercors massif as a function of elevation.





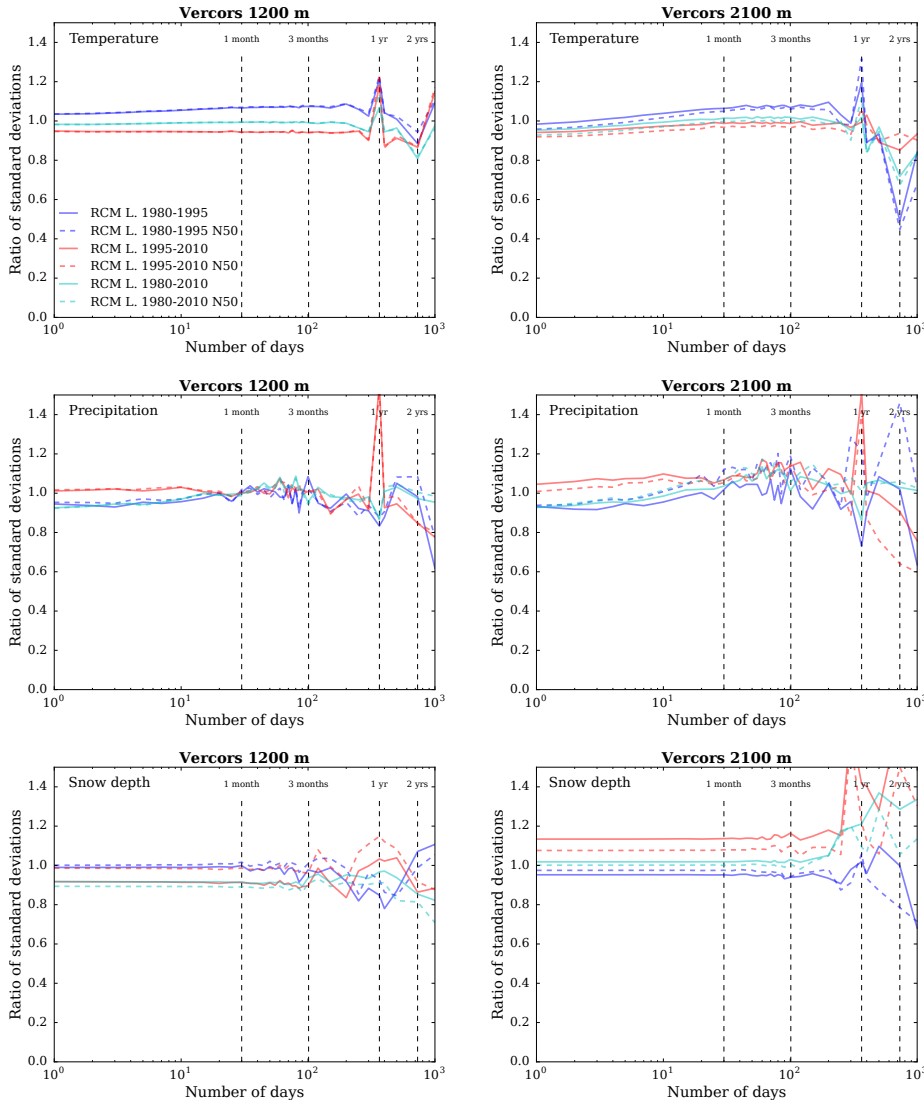

**Figure 7.** Ratio of standard deviations between the SAFRAN reanalysis and adjusted RCM temperature, precipitation and snow depth (using Crocus in this case) as a function of the integration window over the evaluation period, for the Vercors massif at 1200 m a.s.l. and 2100 m a.s.l..




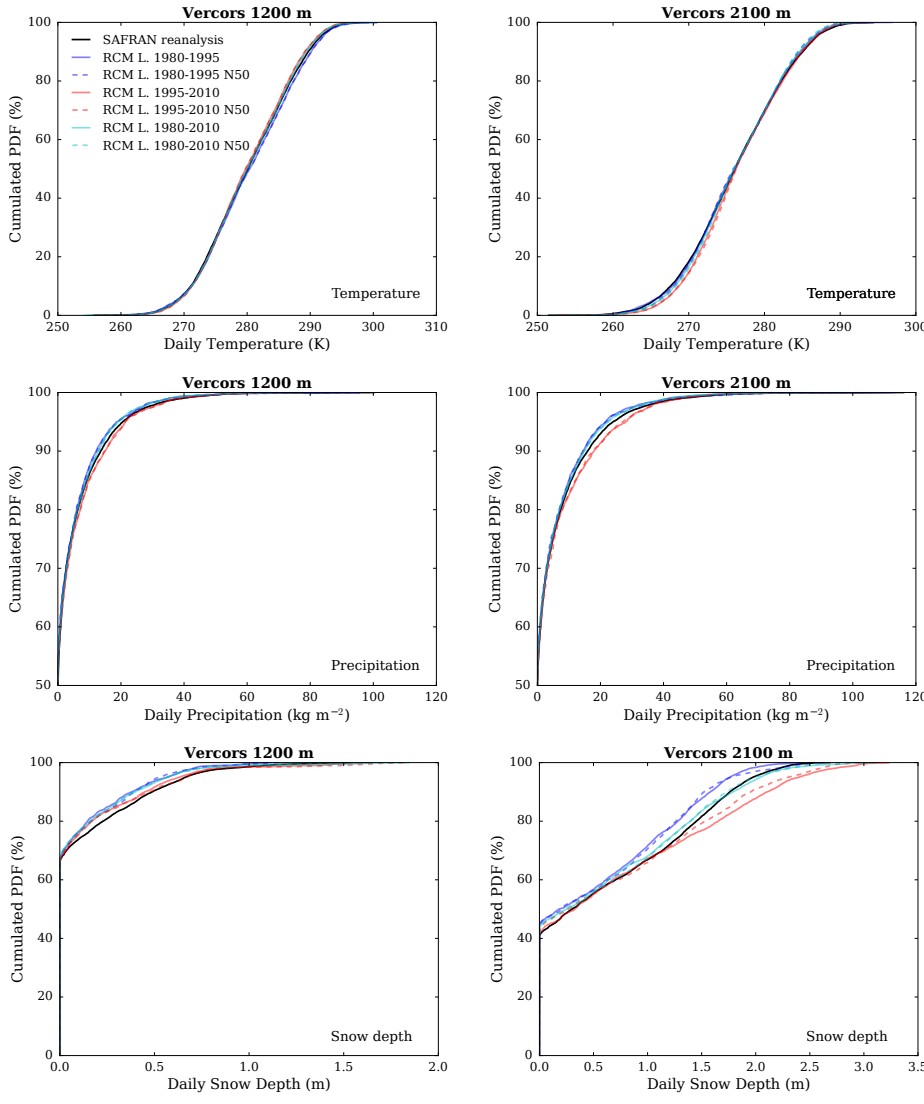

**Figure 8.** Cumulated probability density function (PDF) of daily temperature, precipitation and snow depth (using Crocus in this case) in each adjusted RCM simulation and in the SAFRAN reanalysis over the evaluation period, for the Vercors massif at 1200 m a.s.l. and 2100 m a.s.l..





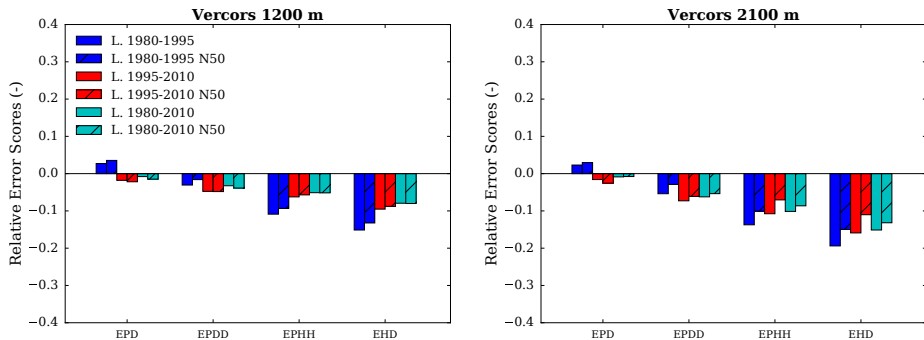

**Figure 9.** Scores for the duration and persistence of precipitation events in each adjusted RCM simulation compared to the SAFRAN reanalysis over the evaluation period, for the Vercors massif at 1200 m a.s.l. and 2100 m a.s.l.. EPD = relative error on the probability of a dry day, EPDD = relative error on the probability of a dry day following a dry day, EPHH = relative error on the probability of a wet day following a wet day, EHD = relative error on the mean duration of wet periods.



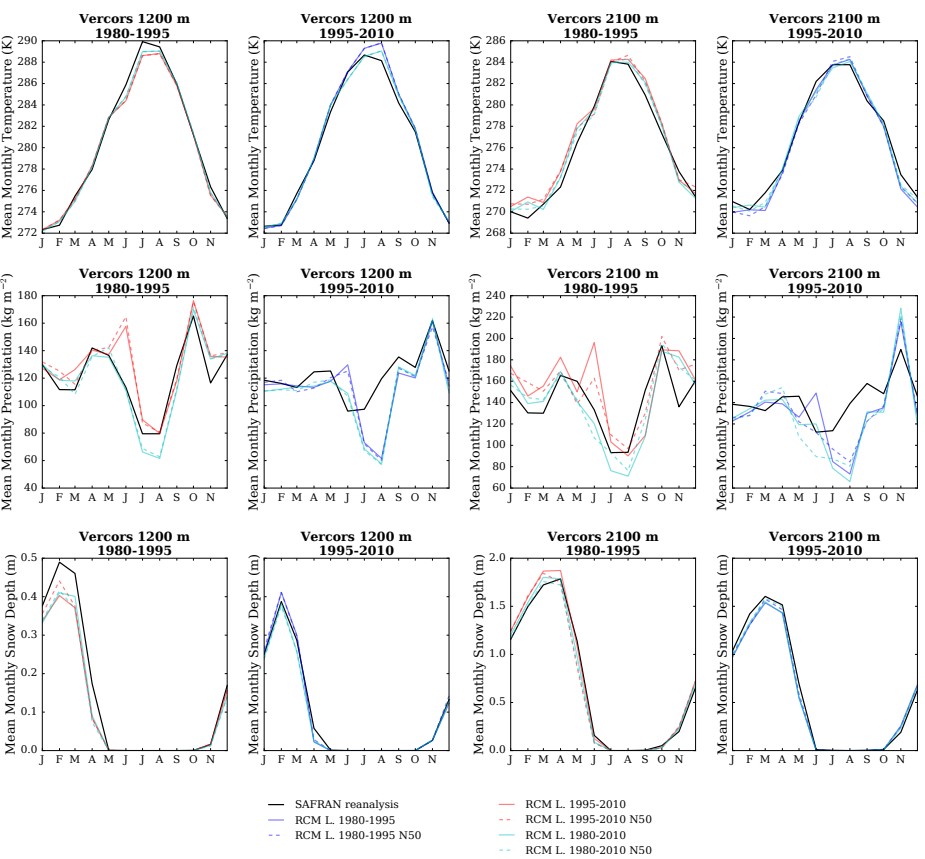

**Figure 10.** Mean annual cycle of temperature, precipitation and snow depth (using Crocus in this case) in each adjusted RCM simulation and in the SAFRAN reanalysis over the period 1980-1995 and 1995-2010, for the Vercors massif at 1200 m a.s.l. and 2100 m a.s.l.. Letters on the x-axis correspond to the different months of the calendar (J = January, F = February, etc.).



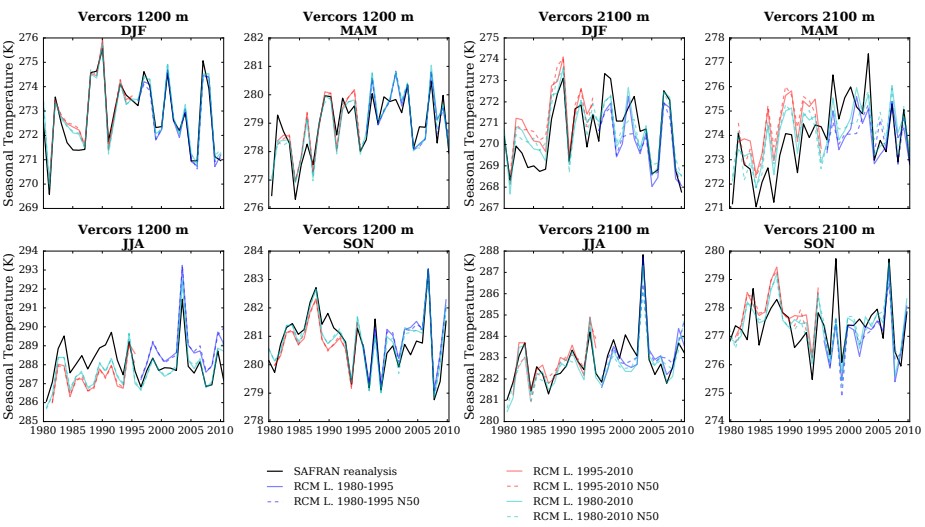

**Figure 11.** Seasonal average time series of temperature from 1980 to 2010 in each adjusted RCM simulation and in the SAFRAN reanalysis, for the Vercors massif at 1200 m a.s.l. and 2100 m a.s.l..

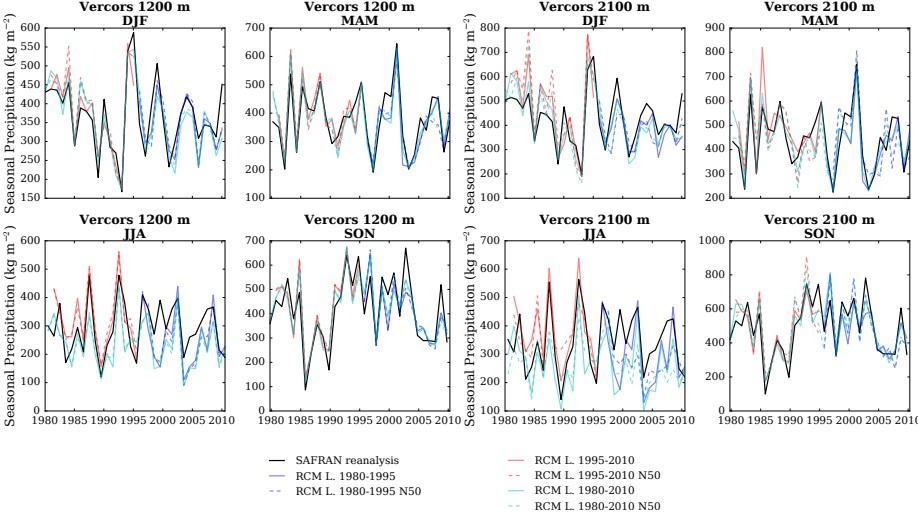

**Figure 12.** Seasonal average time series of precipitation from 1980 to 2010 in each adjusted RCM simulation and in the SAFRAN reanalysis, for the Vercors massif at 1200 m a.s.l. and 2100 m a.s.l..





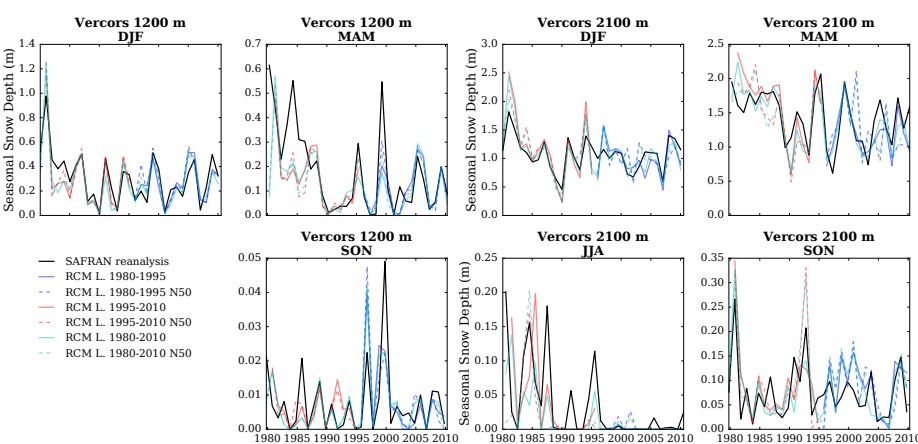

**Figure 13.** Seasonal average time series of snow depth from 1980 to 2010 in each adjusted RCM simulation (used as input for Crocus) and in the SAFRAN/Crocus reanalysis, for the Vercors massif at 1200 m a.s.l. and 2100 m a.s.l..





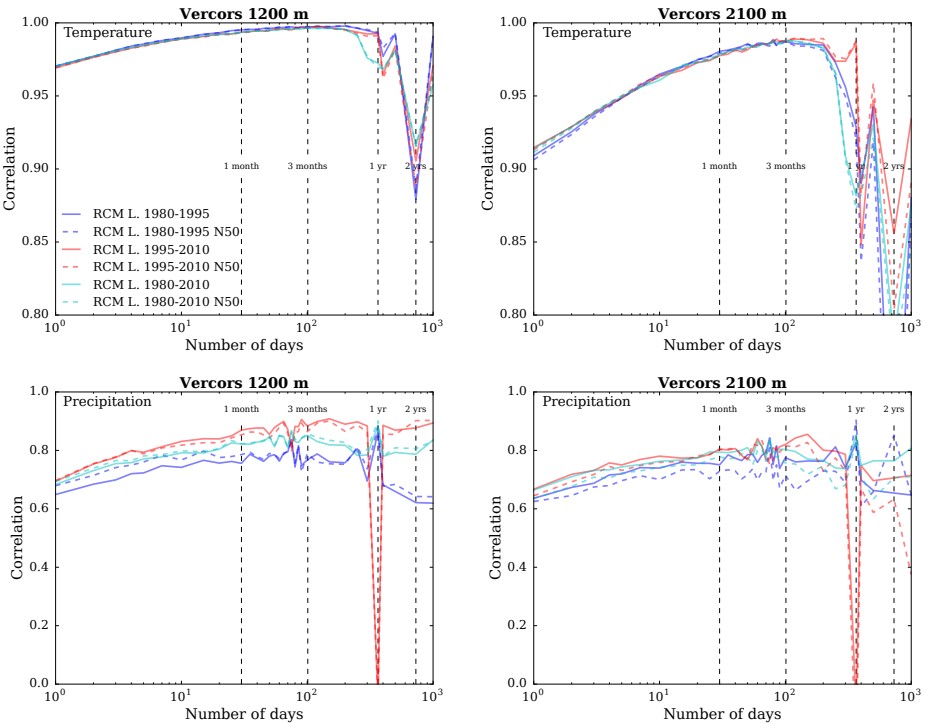

**Figure 14.** Correlation between the SAFRAN reanalysis and adjusted RCM temperature and precipitation as a function of the integration window over the evaluation period, for the Vercors massif at 1200 m a.s.l. and 2100 m a.s.l..

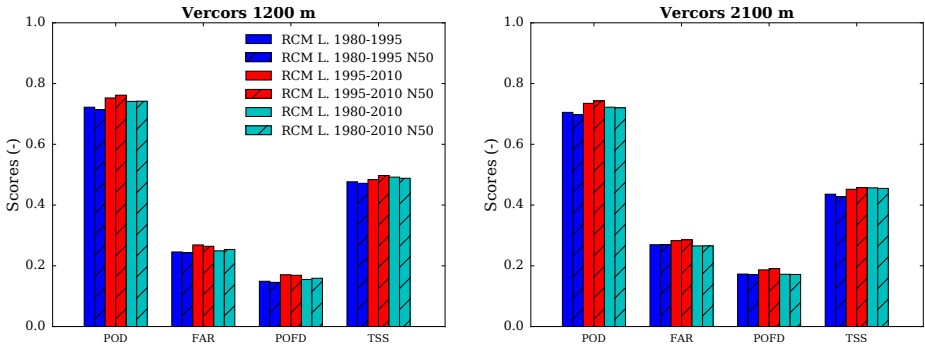

**Figure 15.** Scores for the detection of precipitation events in each adjusted RCM simulation compared to the SAFRAN reanalysis over the evaluation period, for the Vercors massif at 1200 m a.s.l. and 2100 m a.s.l.. POD = probability of detection, FAR = false alarm rate, POFD = probability of false detection, TSS = true skill score.