# Peer review of "The method ADAMONT v1.0 for statistical adjustment of climate projections applicable to energy balance land surface models"

_Geoscientific Model Development, 2017_

## Referee Comment (RC1) · Anonymous Referee #1 · 4 Aug 2017

General comments: This paper describes a new sophisticated method to adjust and disaggregate daily RCM output to hourly values, which are usually necessary to force energy-balanced based land surface models. Such a method is an interesting and useful addition to the field. The work is therefore relevant, although the applied reanalysis data set used as reference is very specific and the performance of the method with other observational datasets first needs to be demonstrated. After the authors investigated the impact of the grid point selection, the "ultimate quantile mapping" and the transferability in time, it would also be interesting to know the impact of the weather regime consideration or not. This is just a wish and since the paper is already long enough I understand if the authors want to cut this point. Therefore, I recommend

publishing the paper once the authors addressed at least the points listed below:

Specific comments:

L80: ADAMONT stands for what?

L123: daily RCM model outputs

L143: 4 weather regimes: Can they be named or described somehow? If I understand right, this means that every day in the reference period has been categorized in one of the four weather types, which is valid for all massifs for this day? Is this already implemented in ADAMONT for Europe as a kind of look-up table? The 4 weather regimes are based on quite old study. What about the consideration of 5 weather regimes as proposed in a recent study (http://onlinelibrary.wiley.com/doi/10.1002/2017GL074188/epdf) ?

L146: Integration: Would "Aggregation" not be the better term?

L148: 6 am to 6 am the next day: Is this UTC or local time?

L150: daily mean: 6 am to 6 am?

L152: Ok, you calculate the 99 percentiles, but what do you mean with "99 percentiles + 0.5 % and 99.5 % quantiles" ?

L160: For RCM values greater than the 99.5 % quantile, a constant adjustment based on the value of this last quantile is applied in order to allow for new extremes.

L170: A further criterion can be applied: Did you apply it or not?

L174: a random draw: This in contradiction to the desired "consecutive time slices" described above!

L175: browsed through: in which direction?

L186: RCM adjusted daily minimum and maximum: It should be mentioned before, that RCM often provide daily minimum and maximum temperatures.

L201: Equation 4: T_h_RCM(24h, i-1) is not available for the first day. What to take then?

L206-209: X_h_SAF should be replaced with X_h_OBS!

L231: Definition of snow year is missing!

L260: all massifs: Should it not be one massif, since the calculation is done by massif?

L282: Replace Method with ADAMONT

L300: I guess a given altitude level means a +/-150 m wide elevation band?

L304-372: References to the corresponding tables and figures would help a lot.

L391-393: Please mention that the good agreement for snow depth is due to the fact that the difference in winter precipitation is small (see Fig. 5)!

L403 & L412: Are there no noteworthy differences between massifs?

L415: smaller than 150 kg m−2 per month: This should also be expressed in percentage!

L422: biases never exceed 50 cm: This should also be expressed in percentage!

L430: Fig. 5 & 6

L533: as found by Lafaysse (2011)

L551: TSS are generally better for massifs of the Northern Alps: Could you please provide some percentage range!

L564: Why not Figs. 10-13?

L575-577: Please give a reference for this statement!

L602: biases for precipitation

L622-623: The new method ADAMONT is able to statistically adjust daily regional

climate model projections and to provide hourly...

L647: ultimate quantile mapping: Should be again explained in more detail for the conclusion section.

Table 2: For a better understanding, the configuration with N=0 should be also labeled as such.

Figure 1: An additional small map with numbered massifs (e.g. right of the elevation color bar) would give the reader a possibility to geographically locate the massifs listed in Table 3, where the same number needs to be inserted.

Figure 3 (top left): Why is the 1800 m elevation not considered?

Figure 3 (top right): I guess the time period of the SAFRAN reanalysis is 1980-2010. Please give this information in the legend or in the figure caption.

Figure 3 (caption L3): different elevations considered (900-2400 m...)

Figure 8: I guess the time period of the SAFRAN reanalysis is 1980-2010. Please give this information in the legend or in the figure caption.

Figure 10: Scale of the y-axis for the two elevations should be the same for comparability. The y-axis labeling in the 2. and the 4. column is missing. Should be like Figure 11.

---

## Referee Comment (RC2) · Anonymous Referee #2 · 15 Aug 2017

Review report for manuscript "The method ADAMONT v1.0 for statistical adjustment of climate projections applicable to energy balance land surface models" by Verfaillie et al. (2017) This study introduces the method ADAMONT v1.0 to adjust and disaggregate daily climate projections from a regional climate model against an observational dataset at hourly time resolution. The method makes use of a refined quantile mapping approach for statistical adjustment and an analogous method for sub-daily disaggregation. The method is capable of producing adjusted hourly time series of temperature, precipitation, wind speed, humidity, and short- and longwave radiation, which can in turn be used to drive any energy balance land surface model (e.g. a fully distributed energy and water balance hydrologic model). The observational dataset used here is

the SAFRAN meteorological reanalysis, which covers the entire French Alps split into 23 massifs, within which meteorological conditions are provided for several 300 m elevation bands. In order to evaluate the skills of the method itself, it is applied to the ALADIN-Climate v5 RCM using the ERA-Interim reanalysis as boundary conditions, for the time period from 1980 to 2010. The authors find the disaggregation method to preserve inter-variable dependency structures although it performed well for temperature compared to precipitation. The manuscript is well organized and the analyses methods are well thought out, except a few points. Please find below a few comments which could help you to improve your manuscript on the way to publication.

Major comment: Line 1 – 64: The authors introduce the need for bias-correction of RCM outputs but completely fail to address the many flaws of bias-adjustment which have been well detailed in Ehret at al 2012: "Should we apply bias correction to global and regional climate model data?" Most impact studies are now utilizing convection permitting models at <4km resolution to overcome some of these limitations. Also, the authors have to specifically state that the results of the quantile mapping are sensitive to data sets used and adjustment method as well. Thus, there is a wide array of uncertainties associated with these kinds of studies.

Minor Comments:

Abstract: I could not tell for which RCP(s) the adjustment was made just by reading the abstract. Please make the abstract a standalone section.

What is "ADAMONT"?

Line 145 – 160: what do you mean by integration? Just use something like "aggregation" for easy understanding. Tmax/Tmin is taken from 6am to 6am? This is not clear at all. When did you take the max and min specifically?

Line 335: The authors should clearly state that the RMSE and mean bias were used to evaluate model performance in terms of reproducing amounts while FAR, POD etc.

for occurrence.
* * *

---

## Author Comment (AC1) · 18 Sep 2017

**Response to Referee 1**

*We thank R1 for this detailed review, which enabled us to significantly improve our article. Enclosed please find a detailed explanation of the revisions we made based on R1's comments. For convenience, comments are in bold and our responses are in italic. Revisions made in the manuscript are presented in italic with grey background.*

**General comments: This paper describes a new sophisticated method to adjust and disaggregate daily RCM output to hourly values, which are usually necessary to force energy-balanced based land surface models. Such a method is an interesting and useful addition to the field. The work is therefore relevant, although the applied reanalysis data set used as reference is very specific and the performance of the method with other observational datasets first needs to be demonstrated. After the authors investigated the impact of the grid point selection, the "ultimate quantile mapping" and the transferability in time, it would also be interesting to know the impact of the weather regime consideration or not. This is just a wish and since the paper is already long enough I understand if the authors want to cut this point. Therefore, I recommend publishing the paper once the authors addressed at least the points listed below:**

*We thank the reviewer for this review, please see our specific responses to each point below.*

*Concerning the remark: « **it would also be interesting to know the impact of the weather regime consideration or not.** », we also think this would be interesting. We haven't looked deeply into this, but as shown in Driouech et al. (2010), the frequency of weather regimes changes in a warmer climate, contributing significantly to the change in precipitation. If we adjust a model irrespective of the regimes, the adjustment may result from a compensation between regime-dependent systematic errors. It is therefore wiser, if sampling permits, to correct this error for each regime separately. As a result, the conditional systematic error for each regime for present climate is, by construction, zero. But since the model regime frequencies are not exactly the same as the observed ones during the training period, the full-sample systematic error is not zero. Our method is thus a compromise : we slightly degrade present climate in oder to expect less bias in future climate. However, in order to keep the manuscript as short as possible, we did not develop this point further.*

**Specific comments:**

**L80: ADAMONT stands for what?**

*ADAMONT is the name of one of the projects which funded this study, which was then used to name the method. We thus prefer to not give any sense to the name chosen.*

**L123: daily RCM model outputs**

*This was included (Line 126).*

**L143: 4 weather regimes: Can they be named or described somehow? If I understand right, this means that every day in the reference period has been categorized in one of the four weather types, which is valid for all massifs for this day? Is this already implemented in ADAMONT for Europe as a kind of look-up table? The 4 weather regimes are based on quite old study. What about the consideration of 5 weather regimes as proposed in a recent study (http://onlinelibrary.wiley.com/doi/10.1002/2017GL074188/epdf) ?**

*There are 4 different regimes defined for each season and it is now explained (Line 144-147) : « RCM weather regimes were determined based on the synoptic fields of the GCM model used as boundary condition for the RCM. In Michelangeli et al. (1995) and Driouech et al. (2010), only regimes for the winter season are defined. We chose to apply the same method to determine weather regimes for the other seasons as well.» For the winter season, regimes have been given a traditional name (the number is arbitrary): 1 = Zonal, 2 = Atlantic Ridge, 3 = Blocking, 4 = Greenland Anticyclone. For other seasons, they don't have any name, we applied the same algorithm as for winter.*

*Indeed, every day in the reference period has been categorized into one of the four weather types for each season, which is valid for all massifs for this day. This is now clarified (Line 142-144) : « The ERA-Interim reanalysis (Dee and Uppala, 2009) was used to infer weather regimes corresponding to each observation date and for all observation points. ».*

*Our choice of regimes is based on the study of Michelangeli et al. (1995), which has served as basis for other studies such as Driouech et al. (2010). Moreover, the initial method has been updated here by computing weather regimes for the different seasons, and by computing the clusters based on ERA-40 (as in previous studies), but using ERA-Interim to infer the weather regime corresponding to each observation date and for all observation points. We could use more regimes, but this would endanger the robustness of our results, because of the too limited number of data used to infer quantile values if too many weather regimes are considered, some corresponding to a small number of days. Four regimes is found to be satisfying for this study, as it ensures a sufficiently large size of the datasets for quantile mapping. We have rephrased the end of the paragraph (Line 149-153) : « This number is a compromise between accuracy of the correction and robustness of the percentile estimation (more regimes can be used, such as in Ummenhofer et al. (2017)). On the other hand this relatively small number of regimes ensures a sufficiently large size of the datasets used for quantile mapping (which are, as described below, further partitioned into 4 seasons DJF, MAM, JJA, SON). »*

*A new Figure was added to represent the different regimes used (Line 153-154 & Fig. 1) : « Figure 1 represents the different regimes used in this study. »*

**L146: Integration: Would "Aggregation" not be the better term?**

*Yes, this would be better. We changed the word « integration » to « aggregation » and « integrated » to « aggregated » (Line 155 and caption of Table 1).*

**L148: 6 am to 6 am the next day: Is this UTC or local time?**

*This is UTC. We have now changed the sentence to introduce this precision (Line 157): « (from 6:00 UTC to 6:00 UTC the next day) ».*

**L150: daily mean: 6 am to 6 am?**

*No, this is UTC. This was included (Line 159-161).*

**L152: Ok, you calculate the 99 percentiles, but what do you mean with "99 percentiles + 0.5 % and 99.5 % quantiles" ?**

*The percentiles (1 %, 2 %, …, 99%) are calculated, and we also calculate the 0.5 % quantile and the 99.5 % quantile. This sentence was changed to (Line 162-163):* « *The quantile values (the 99 percentile values as well as the 0.5 % and 99.5 % quantile values)...* »

**L160: For RCM values greater than the 99.5 % quantile, a constant adjustment based on the value of this last quantile is applied in order to allow for new extremes.**

*This precision was added (Line 170-172).*

**L170: A further criterion can be applied: Did you apply it or not?**

*Yes, this sentence was corrected (Line 181) :* « *A further criterion is applied...* ».

**L174: a random draw: This in contradiction to the desired "consecutive time slices" described above!**

*R1 is right. This point was not clear enough. We have rephrased the paragraph as follows (Line 185-191) :* « *For the first RCM date, a random draw amongst all available observational dates is performed, then the dates are browsed through chronologically until one meets all the requirements outlined above. This analogous day is then used in the following step for all variables. If the following analogue day in the observations still meets all requirements, it is selected as analogue for the following day in the RCM (to ensure as far as possible consecutive time slices). A new random draw is only performed once the analogue fails to meet all requirements described above.* »

**L175: browsed through: in which direction?**

*They are browsed through chronologically. This is now specified (Line 186) :* « *(…) then the dates are browsed through chronologically until one meets all the requirements outlined above.* »

**L186: RCM adjusted daily minimum and maximum: It should be mentioned before, that RCM often provide daily minimum and maximum temperatures.**

*This is now mentioned in point n°3 (Line 156-158) :* « *for temperature, the daily minimum and maximum values (from 6:00 UTC to 6:00 UTC the next day) are selected (RCMs generally offer daily minimum and maximum temperature).* »

**L201: Equation 4: T_h_RCM(24h, i-1) is not available for the first day. What to take then?**

*We thank R1 for this remark. Indeed, this is a point that we did not describe in detail in the article. This is now included (Line 218-224) :* « *For specific cases, i.e. for the first day where* $T_{RCM}$ *(24h, i − 1) does not exist, or if the determinant of our system is too close to zero (< 0.1), or in the case where a < 0, a much simpler equation is used, in which we only ensure that final mimimum and maximum daily values correspond to the RCM adjusted minimum and maximum values, by solving:*

$$a = (Tmax_{RCM}^{d,adj}(i) - Tmin_{RCM}^{d,adj}(i)) / ((Tmax_{OBS}^{h} - Tmin_{OBS}^{h})$$
$$b = Tmax_{RCM}^{d,adj}(i) - a\ Tmax_{OBS}^{h}.\ »$$

**L206-209: X_h_SAF should be replaced with X_h_OBS!**

*It is now corrected (Line 229-230).*

**L231: Definition of snow year is missing!**

*This is now defined (Line 251-253) : « The resulting adjusted hourly time series for each variable are obtained for each snow year (from the 1st of August to the 31st of July of the following year) ».*

**L260: all massifs: Should it not be one massif, since the calculation is done by massif?**

*No, this criterion on the wet/dry analogue days is applied to all massifs in order to « maximise the consistency between massifs after the adjustment process », as indicated in the text. Please keep in mind, however, that this criterion is a second order criterion for the selection of analogous days, the first order criterions being the month of the year, the weather regime and whenever possible, consecutive time slices for consecutive RCM dates.*

**L282: Replace Method with ADAMONT**

*The section title was changed to « ADAMONT method evaluation » (Line 306).*

**L300: I guess a given altitude level means a +/-150 m wide elevation band?**

*We have added a clearer description of elevation bands earlier in the article (Line 267-269) : «SAFRAN data are available for elevation bands with a resolution of 300 m, i.e. altitude levels 600, 900, 1200, 1500 m etc. are typically considered, making it possible to extract meteorological information at these altitude levels, or in-between using altitude interpolation. »*

**L304-372: References to the corresponding tables and figures would help a lot.**

*The reviewer is correct that it could ease reading, but given that Tables and Figures are introduced in the Results section, referring to them earlier in the manuscript would recquire major changes to their description. Indeed, this would alter Figures and Tables order, and lead to the need to introduce them fully in the Methods sections, before the introduction of their detailed content, which is a problem too. Given these condiserations, we chose to not introduce the Tables and Figures there.*

**L391-393: Please mention that the good agreement for snow depth is due to the fact that the difference in winter precipitation is small (see Fig. 5)!**

*This is now indicated (Line 441-444) : « Smaller autumn and winter precipitation biases lead to a good agreement between the magnitude of average snow depth in the different adjusted RCM simulations and the results obtained using the reanalysis as meteorological input (as noted in Fig. 4). »*

**L403 & L412: Are there no noteworthy differences between massifs?**

*The large biases and RMSEs values obtained when using raw RCM simulations compared to adjusted simulations are features common to all massifs. It is now more clearly indicated (Line 426-428) : « This highlights the large biases and RMSEs values obtained when using raw RCM simulations compared to adjusted simulations, a feature common to all massifs (Figs. 5-6 and Supplementary Information)». So is the fact that the longer learning period 1980-2010 generally presents smaller biases and RMSEs. The word « generally » in this sense encompasses the analysis across all massifs.*

**L415: smaller than 150 kg m−2 per month: This should also be expressed in percentage!**

*OK. This information was added (Line 439-441) :* « *Biases of the adjusted simulations remain smaller than 150 kg m $^{-2}$ per month in absolute value, corresponding to up to 90% depending on the massif and altitude* »

**L422: biases never exceed 50 cm: This should also be expressed in percentage!**

*This was added (Line 450-451) :* « *For snow depth, the biases never exceed 50 cm, which corresponds to up to 50% depending on the altitude and the massif* »

**L430: Fig. 5 & 6**

*Indeed (it is now Figs. 6-7). This was corrected (Line 460-461).*

**L533: as found by Lafaysse (2011)**

*Done (Line 562).*

**L551: TSS are generally better for massifs of the Northern Alps: Could you please provide some percentage range!**

*This information was added (Line 580-582) :* « *TSS are generally better for massifs of the Northern Alps (0.25 to 0.6) than the Southern Alps (0.1 to 0.4, Supplementary Information)* ».

**L564: Why not Figs. 10-13?**

*Yes (now Figs. 11-14), this error was corrected (Line 593).*

**L575-577: Please give a reference for this statement!**

*This was added (Line 603-604) :* « *(…) when most observations from mountain stations are not available (Gobiet et al., 2015)* ».

**L602: biases for precipitation**

*This was included (Line 630) .*

**L622-623: The new method ADAMONT is able to statistically adjust daily regional climate model projections and to provide hourly. . .**

*This sentence was rephrased accordingly (Line 651-653).*

**L647: ultimate quantile mapping: Should be again explained in more detail for the conclusion section.**

*We thank R1 for this suggestion. This was added (Line 676-679) :* « *the ultimate quantile mapping applied to snowfall and rainfall (i.e., after a first quantile mapping on total precipitation, an additional quantile mapping against the observational dataset is applied for daily cumulated adjusted RCM rainfall and snowfall separately)* »

**Table 2: For a better understanding, the configuration with N=0 should be also labeled as such.**

*OK. This has now been included in Table 2, Figs 4-16 and in the Supplementary Information.*

**Figure 1: An additional small map with numbered massifs (e.g. right of the elevation color bar) would give the reader a possibility to geographically locate the massifs listed in Table 3, where the same number needs to be inserted.**

*This is now included in Fig. 3 and Table 3.*

**Figure 3 (top left): Why is the 1800 m elevation not considered?**

*It is considered, but it corresponds to the same ALADIN RCM grid point as for 1500 m, so we don't see the line corresponding to 1800 m. We included the following sentence in the caption of Fig. 4 to indicate this : « In this case the 1500 m and 1800 m lines are similar. »*

**Figure 3 (top right): I guess the time period of the SAFRAN reanalysis is 1980-2010. Please give this information in the legend or in the figure caption.**

*This is now included in the figure caption (Fig. 4).*

**Figure 3 (caption L3): different elevations considered (900-2400 m. . .)**

*Done (Fig. 4).*

**Figure 8: I guess the time period of the SAFRAN reanalysis is 1980-2010. Please give this information in the legend or in the figure caption.**

*This is now included in the figure caption (Fig. 9).*

**Figure 10: Scale of the y-axis for the two elevations should be the same for comparability. The y-axis labeling in the 2. and the 4. column is missing. Should be like Figure 11.**

*We are afraid that if we use the same scale for the y-axis for 1200 m and 2100 m, some curves for 1200 m won't be readable anymore. The values at 1200 m and 2100 m can be very different, especially for precipitation and even more for snow depth. This is why we would prefer to keep the scales for the y-axis as is. However, we have changed the labeling of the y-axis as proposed by R1 (Fig. 11).*

---

## Author Comment (AC2) · 18 Sep 2017

**Response to Referee 2**

*We thank R2 for this helpful review. Enclosed please find a detailed explanation of the revisions we made based on R2's comments. For convenience, comments are in bold and our responses are in italic. Revisions made in the manuscript are presented in italic with grey background..*

**Review report for manuscript "The method ADAMONT v1.0 for statistical adjustment of climate projections applicable to energy balance land surface models" by Verfaillie et al. (2017) This study introduces the method ADAMONT v1.0 to adjust and disaggregate daily climate projections from a regional climate model against an observational dataset at hourly time resolution. The method makes use of a refined quantile mapping approach for statistical adjustment and an analogous method for sub-daily disaggregation. The method is capable of producing adjusted hourly time series of temperature, precipitation, wind speed, humidity, and short- and longwave radiation, which can in turn be used to drive any energy balance land surface model (e.g. a fully distributed energy and water balance hydrologic model). The observational dataset used here is the SAFRAN meteorological reanalysis, which covers the entire French Alps split into 23 massifs, within which meteorological conditions are provided for several 300 m elevation bands. In order to evaluate the skills of the method itself, it is applied to the ALADIN-Climate v5 RCM using the ERA-Interim reanalysis as boundary conditions, for the time period from 1980 to 2010. The authors find the disaggregation method to preserve inter-variable dependency structures although it performed well for temperature compared to precipitation. The manuscript is well organized and the analyses methods are well thought out, except a few points. Please find below a few comments which could help you to improve your manuscript on the way to publication.**

*We thank the reviewer for this review, please see our specific responses to each point below.*

**Major comment: Line 1 – 64: The authors introduce the need for bias-correction of RCM outputs but completely fail to address the many flaws of bias-adjustment which have been well detailed in Ehret at al 2012: "Should we apply bias correction to global and regional climate model data?" Most impact studies are now utilizing convection permitting models at <4km resolution to overcome some of these limitations. Also, the authors have to specifically state that the results of the quantile mapping are sensitive to data sets used and adjustment method as well. Thus, there is a wide array of uncertainties associated with these kinds of studies.**

*The reviewer is correct that bias-correction is not a perfect solution, but it is still a necessary step when using regional climate model data for impact studies (Maraun 2016), be it convection permitting or not. In addition, while a few studies have recently emerged using non-hydrostatic high-resolution model approaches targeting summertime processes such as convection-driven events (e.g. Ban et al., 2015, Giorgi et al., 2016, https://www.hymex.org/cordexfps-convection/wiki/doku.php?id=modellist ), in some areas impact studies have only marginally employed such models and most existing studies extensively rely on 10-km resolution regional climate models such as those employed in EURO-CORDEX. For example, studies addressing snow in mountainous areas have only in a few cases employed high resolution non-hydrostastic models (e.g. Musselmann et al., 2017), mostly for upstream research and process studies rather than for impact studies, which require very low biases because of the threshold effects at play in snowpack processes. We therefore believe that, even though future studies will increasingly*

*employ high resolution convection permitting regional climate models, many impact studies will be carried out using hydrostatic models as part of large-scale projects such as EURO-CORDEX and beyond. Furthermore, as indicated above, convection-permitting models are not immune of biases (Prein et al., 2015) and will require appropriate adjustment for being used in impact assessments. Concerning the sensitivity of quantile mapping to the data sets used and adjustment method, we have now added the following sentence to account for this (Line 64-66) : « Furthermore, the performance level of quantile mapping methods is sensitive to the observation data set used and the detailed characteristics of their implementation, which requires specific attention. »*

*Ban, N., J. Schmidli and C. Schär, 2015: Heavy precipitation in a changing climate: Does short-term summer precipitation increase faster?. Geophys. Res. Lett., 42, 1165-1172.*

*Giorgi F., C. Torma, E. Coppola, N. Ban, C. Schär and S. Somot, 2016: Enhanced summer convective rainfall at Alpine high elevations in response to climate warming, Nature Geoscience, 9, 584-590.*

*Musselman, K.N, M.P. Clark, C. Liu, K. Ikeda, and R. Rasmussen, 2017:  Slower snowmelt in a warmer world, Nature Climate Change, 7, 214-219.*

*Prein, A.F., W. Langhans, G. Fosser, A. Ferrone, N. Ban, K. Goergen, M. Keller, M. Tölle, O. Gutjahr, F. Feser, E. Brisson, S. Kollet, J. Schmidli, N.P.M. van Lipzig, and R. Leung, 2015: A review on regional convection-permitting climate modeling: Demonstrations, prospects, and challenges, Rev. Geophys., 53, 323-361.*

**Minor Comments:**

**Abstract: I could not tell for which RCP(s) the adjustment was made just by reading the abstract. Please make the abstract a standalone section.**

*No RCP was used. In this article, we only focus on the evaluation for the recent period 1980-2010, as indicated in the abstract (Line 11-13) : « In order to evaluate the skills of the method itself, it is applied to the ALADIN-Climate v5 RCM using the ERA-Interim reanalysis as boundary conditions, for the time period from 1980 to 2010. »*

**What is "ADAMONT"?**

*ADAMONT is the name of one of the projects which funded this study. There is no meaningful definition beyond this name.*

**Line 145 – 160: what do you mean by integration? Just use something like "aggregation" for easy understanding. Tmax/Tmin is taken from 6am to 6am? This is not clear at all. When did you take the max and min specifically?**

*We thank R2 for this remark.*
*We changed the word « integration » to « aggregation » and « integrated » to « aggregated » (Line 155 and caption of Table 1). Maximum and minimum values are calculated from 6 am to 6 am, and only for temperature. For other variables, the daily mean (from 6 am to 6 am, this information has now been included) or the last value of each day is used.*
*We have slightly changed this paragraph to make it clearer (Line 156-161) : « for temperature, the daily minimum and maximum values (from 6:00 UTC to 6:00 UTC the next day) are selected (RCMs generally offer daily minimum and maximum temperature). For wind speed and humidity, the last value of each day (at 6:00 UTC) is selected (in order to be comparable to an instantaneous value), and for precipitation and radiation, the daily mean (6:00 UTC to 6:00 UTC) is used. »*

**Line 335: The authors should clearly state that the RMSE and mean bias were used to evaluate model performance in terms of reproducing amounts while FAR, POD etc. for occurrence.**

*OK, even though we don't evaluate model performances, but rather the performances of the ADAMONT method.*

*This is now stated (Line 359-362) : « – the root mean square error (RMSE) and the mean bias over the evaluation period, computed over seasonal integration periods based on the SAFRAN and the adjusted RCM datasets (to evaluate the method performance in terms of reproducing amounts);*

*– scores specific to the detection of occurrence of precipitation events (...) »*